# Robust Multimodal Learning via Cross-Modal Proxy Tokens

**Md Kaykobad Reza**
*University of California Riverside*                                          mreza025@ucr.edu

**Ameya Patil**
*Amazon*                                                                      ameyapat@amazon.com

**Mashhour Solh**
*Amazon*                                                                      mashhour@amazon.com

**M. Salman Asif**
*University of California Riverside*                                          sasif@ucr.edu

**Reviewed on OpenReview:** *https://openreview.net/forum?id=Wtc6wvcYJO*

## Abstract

Multimodal models often experience a significant performance drop when one or more modalities are missing during inference. To address this challenge, we propose a simple yet effective approach that enhances robustness to missing modalities while maintaining strong performance when all modalities are available. Our method introduces cross-modal proxy tokens (CMPTs), which approximate the class token of a missing modality by attending only to the tokens of the available modality without requiring explicit modality generation or auxiliary networks. To efficiently learn these approximations with minimal computational overhead, we employ low-rank adapters in frozen unimodal encoders and jointly optimize an alignment loss with a task-specific loss. Extensive experiments on five multimodal datasets show that our method outperforms state-of-the-art baselines across various missing rates while achieving competitive results in complete-modality settings. Overall, our method offers a flexible and efficient solution for robust multimodal learning. The code for this paper is available at: https://github.com/CSIPlab/Cross-Modal-Proxy-Tokens.

## 1 Introduction

Multimodal learning (Ngiam et al., 2011; Xu et al., 2023) integrates information from multiple (diverse) input sources to improve the performance on downstream tasks. For example, examining a movie poster image along with a short text synopsis can provide a greater insight into the movie's type and genre. In recent years, a number of methods have been proposed to effectively combine information from multiple modalities, including early-stage fusion (Mo & Morgado, 2024), token-level fusion (Wang et al., 2022b), channel exchange (Wang et al., 2020b), and bottleneck mid-fusion (Nagrani et al., 2021a). Furthermore, specialized architectures have been proposed for specific multimodal applications, such as vision-language tasks (Kim et al., 2021; Li et al., 2019; Lu et al., 2019), action recognition (Woo et al., 2023; Zhou et al., 2022), and segmentation (Zhang et al., 2023a; Reza et al., 2024b). Most of these approaches assume that all modalities are available for every sample during both training and inference, yielding better performance than unimodal baselines. However, their performance drops sharply when one or more modalities get missing (Reza et al., 2024a; Ma et al., 2022).

In practical applications, input modalities can be missing during training and/or inference for various reasons; such as lack of synchronous data capture (limited paired data during training) and sensor failure or privacy concerns (modalities missing during inference). Ma et al. (2022) observed that multimodal transformers such as ViLT (Kim et al., 2021) exhibit a drastic performance drop when one or more modalities are missing

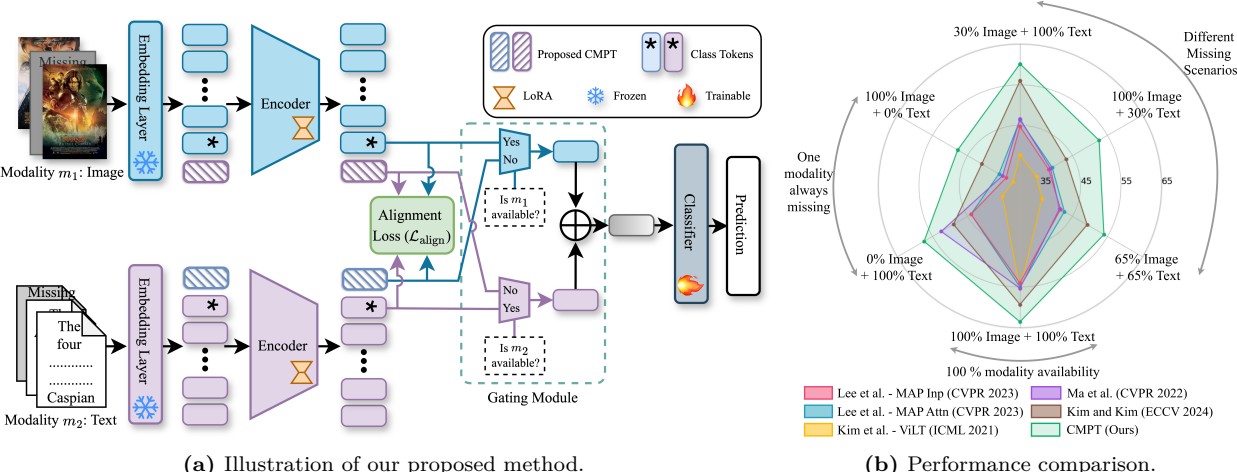

**(a)** Illustration of our proposed method.

**(b)** Performance comparison.

**Figure 1:** (a) We introduce Cross-Modal Proxy Tokens (CMPTs), a novel approach to address missing modality challenges. CMPTs effectively learn to approximate missing modality class tokens by adapting pretrained encoders through a joint optimization of alignment and task-specific objectives. Our approach accommodates both complete and missing modalities during training and inference, thereby enhancing robustness across varying missing modality scenarios. (b) CMPTs achieve state-of-the-art performance, consistently outperforming recent baseline methods in both complete and missing modality scenarios. Following the experimental setup of Lee et al. (2023); Kim & Kim (2024), the radar plot illustrates F1-macro scores on the MM-IMDb dataset across varying modality availability.

during inference as shown in Figure 1b (*yellow* curve). They developed an optimal fusion strategy to recover the performance loss of multimodal transformers in the missing modality scenarios (*purple* curve). Subsequently, Lee et al. (2023) developed a prompt tuning technique (*pink* and *blue* curves), and Kim & Kim (2024) combined the prompt-based approach with Variance Covariance Invariance regularization (VICReg) (*brown* curve) to further improve the robustness of multimodal models to missing modalities. Despite these efforts, achieving consistently strong performance under varying degrees of missing modalities remains an open challenge. In this work, we aim to improve robustness across a wide range of missing modality conditions while maintaining strong performance when all modalities are available.

In this paper, we introduce a simple yet effective framework to *approximate* missing modalities from the available ones for robust multimodal learning. Specifically, we introduce **cross-modal proxy tokens (CMPTs)**, which learn to approximate the class tokens of missing modalities by attending to all tokens of the available modality, enabling effective handling of various missing modality scenarios. As illustrated in Figure 1a, our framework integrates pre-trained unimodal encoders into a multimodal learning setting while introducing CMPTs to partially bridge the performance gap in missing modality scenarios. To ensure efficient learning of CMPTs with minimal additional learnable parameters, we employ low-rank adapters within the frozen unimodal encoders and jointly optimize both an alignment loss and a task-specific loss. The radar plot in Figure 1b demonstrates the superior performance of our approach compared to five existing methods (Kim et al., 2021; Lee et al., 2023; Kim & Kim, 2024; Ma et al., 2022) across six different complete and missing modality scenarios on the MM-IMDb dataset. Our CMPT-based model consistently outperforms all baselines across all the scenarios. We conduct extensive experiments on five multimodal datasets, comparing our approach against existing state-of-the-art baselines. Results show that our framework achieves state-of-the-art performance in both missing modality (Section 4.3) and complete modality (Section 4.4) scenarios, while maintaining a minimal number of learnable parameters. Ablation studies further confirm that our method significantly improves performance even in severe missing-modality scenarios and provides performance enhancement across most of the classes, leading to an overall performance boost (Section 4.6). Main contributions of our work are as follows.

- We propose CMPTs to effectively learn an *approximation* of the missing modality from the available modality with minimal adaption of pre-trained unimodal encoders via an alignment loss.

- Our approach outperforms 12 recent baseline methods across four multimodal datasets in various missing-modality scenarios, demonstrating state-of-the-art performance (Section 4.3).

- Across five datasets, our method achieves on-par or superior performance compared to state-of-the-art techniques when all modalities are present during training and inference (Section 4.4).

- We demonstrate that CMPTs generalize effectively to unseen missing rates (Section 4.5) and boost performance in most of the classes (Section 4.6).

## 2  Related Work

Recent works (Ma et al., 2022; Hazarika et al., 2022; Reza et al., 2024a) show that performance can degrade significantly when modalities are missing, as multimodal models are not inherently robust to such scenarios. Several approaches have been proposed to address this issue. In this section, we primarily discuss works where the main focus is on making multimodal models robust to missing modalities. More broader related work on multimodal learning is discussed in appendix Section A1.

**Robust model design** is one of the approaches for handling missing modalities. Recently, Shin et al. (2024) developed a robust framework utilizing modality masking and knowledge distillation. Wang et al. (2023) proposed ShaSpec to impute missing features through modality-specific and modality-shared feature learning. Wang et al. (2022a) proposed TokenFusion, which dynamically replaces uninformative tokens to enhance robustness. Additionally, different fusion strategies (Choi & Lee, 2019; Fan et al., 2023a; Lin et al., 2023) have been developed to improve robustness across various missing-modality scenarios. However, many of these approaches are task-specific, limiting their generalization to other multimodal tasks. In our work, we focus on reusing pre-trained unimodal encoders, making it flexible for extension to different input modalities and multimodal tasks.

**Robust training and model adaptation** can enhance missing modality robustness. Recently, Reza et al. (2024a) proposed a parameter-efficient adaptation framework for different multimodal tasks. Shi et al. (2024) introduced a modality-aware low-rank adaptation method (MoRA) that uses a shared down-projection layer to map inputs to a low-dimensional space, and then applies modality-specific up-projections for missing modality adaptation. Nezakati et al. (2024) trained a single model to infer missing information by projecting available modalities. Ma et al. (2021) introduced SMIL, a Bayesian meta-learning method for flexible handling of missing modalities during both training and testing. Ma et al. (2022) proposed a multi-task optimization approach with a differentiable algorithm for optimal fusion strategy selection. Maheshwari et al. (2024) developed a semi-supervised framework that leverages unlabeled data to improve robustness, while Wei et al. (2023) used a teacher network to transfer multimodal information for improved missing modality robustness. These methods, however, either require extensive adaptation or specialized training procedures. In our approach, we employ lightweight adaptation of unimodal encoders (using Rank-1 LoRA adapters) and a single extra loss term to explicitly learn cross-modal relationships. Hence, the overall training flow is mostly left unchanged.

**Prompt tuning** methods improve missing modality performance by incorporating learnable prompts across multiple layers of the model. Lee et al. (2023) introduced missing-aware prompts that learns a separate set of prompts for each modality combination. Later, Jang et al. (2024) simplified the approach, showing that a single set of prompts per modality can achieve similar performance. More recently, Dai et al. (2024) proposed a multi-step adaptive prompt learning method through modality alignment, while Kim & Kim (2024) introduced a read-only prompt tuning approach with Variance Covariance Invariance regularization. Although effective, these methods require learning and storing a large number of prompts, typically 16 prompts (Lee et al., 2023; Jang et al., 2024) per modality per layer, for each missing modality scenario. In contrast, our method uses *a single cross-modal proxy token* per modality to estimate the missing modality feature from the available modality, thereby reducing the complexity.

We carry out extensive experimental comparisons with all the above prompt-based techniques to demonstrate superior performance across various missing modality scenarios. Furthermore, our approach is efficient, uses few learnable parameters, and is robust to missing modalities in both training and testing which we discuss in Section 4.

# 3 Proposed Approach

In this paper, we propose a new framework to train multimodal systems such that they retain strong performance under different scenarios of missing modalities during inference. Furthermore, we seek to develop a simple and efficient framework that can be used for different multimodal tasks and datasets without any specialized (task-specific) changes. To achieve these objectives, we propose **cross-modal proxy tokens (CMPTs)**, which effectively learn to approximate the class tokens of missing modalities by adapting pretrained, modality-specific (transformer-based) encoders through a joint optimization of alignment and task-specific objectives, as illustrated in Figure 1a.

## 3.1 Multimodal embedding tokens

Let us consider a multimodal dataset containing two modalities $\mathcal{M} = \{m_1, m_2\}$. We denote the available dataset as $\mathcal{D} = \{\mathcal{D}_c, \mathcal{D}_{m_1}, \mathcal{D}_{m_2}\}$, which can be further divided into three subsets: $\mathcal{D}_c = \{x_{m_1}, x_{m_2}, y\}$ where both modalities are present with corresponding output labels $y$; $\mathcal{D}_{m_1} = \{x_{m_1}, \varnothing, y\}$ where only modality $m_1$ $(x_{m_1})$ is available with corresponding labels $y$; $\mathcal{D}_{m_2} = \{\varnothing, x_{m_2}, y\}$ where only modality $m_2$ $(x_{m_2})$ is available with corresponding labels $y$. Here, $\varnothing$ denotes a placeholder for the missing modality (e.g., empty string for text, zeros for image, audio, or video) following prior works (Lee et al., 2023; Reza et al., 2024a).

For each input modality, we assume that a transformer-based encoder, pre-trained on some large dataset, is available. This is a common and effective practice in modern multimodal learning, particularly given the wide variety of high-performing, pre-trained models now readily accessible (Devlin et al., 2019; Dosovitskiy et al., 2021; Gong et al., 2021). Leveraging these modality-specific encoders trained on extensive data leads to improved performance and faster convergence (Han et al., 2023; Xu et al., 2023). Specifically, each modality $m \in \mathcal{M}$ is processed through a pre-trained embedding layer, EmbeddingLayer$_m$, with parameters $\Theta_{e,m}$. The input $x_m$ for each modality is passed through its corresponding embedding layer to generate modality-specific tokens as

$$\mathcal{T}_m = \text{EmbeddingLayer}_m(x_m; \Theta_{e,m}), \tag{1}$$

where $\mathcal{T}_m \in \mathbb{R}^{N \times d}$ denotes $N$ tokens, each of dimension $d$, for modality $m$. During training, we keep the pretrained parameters $\Theta_{e,m}$ frozen.

## 3.2 Cross-Modal Proxy Tokens (CMPTs)

To compensate for the missing modalities, we introduce cross-modal proxy tokens (CMPTs) that *approximate* the *missing* modality by attending only to the tokens of the *available* modality. For each modality $m \in \mathcal{M}$, we concatenate one CMPT ($\mathcal{T}_{\text{CMPT},m} \in \mathbb{R}^{1 \times d}$) with the class token (CLS) ($\mathcal{T}_{\text{CLS},m} \in \mathbb{R}^{1 \times d}$) and all the modality specific tokens $\mathcal{T}_m$ to generate the modified embedding tokens $\tilde{\mathcal{T}}_m \in \mathbb{R}^{(N+2) \times d}$ as

$$\tilde{\mathcal{T}}_m = \text{Concat}(\{\mathcal{T}_{\text{CMPT},m}, \mathcal{T}_{\text{CLS},m}, \mathcal{T}_m\}). \tag{2}$$

Note that the proposed CMPT serves as an extra token, similar to the class token (Dosovitskiy et al., 2021; Devlin et al., 2019; Gong et al., 2021), but with a complementary purpose. Class token attends to all the tokens of a given modality to aggregate information and learn a rich feature representation of that modality. In contrast, CMPT attends to all the tokens of a given modality to learn an *approximation* of the class token of the other modality (e.g., $\mathcal{T}_{\text{CMPT},m_1}$ approximates $\mathcal{T}_{\text{CLS},m_2}$ while attending to $\mathcal{T}_{m_1}$).

## 3.3 Learn CMPTs by alignment and adaptation

For each modality, we assume that a pretrained transformer encoder $f_m$ with parameters $\Theta_{f,m}$ is available. The encoder for each modality needs to be updated so that they can learn the corresponding CMPTs. To keep the number of trainable parameters minimal, we use low-rank adaptation (LoRA) (Hu et al., 2022) to learn a small number of parameters $\Delta_{f,m}$. The modified embedding tokens for modality $m$, $\tilde{\mathcal{T}}_m$, are then passed through the adapted encoder for modality $m$ to get the final output tokens as

$$\hat{\mathcal{T}}_m = f_m(\tilde{\mathcal{T}}_m; \Theta_{f,m}, \Delta_{f,m}), \tag{3}$$

where $\hat{\mathcal{T}}_m \in \mathbb{R}^{(N+2)\times d}$ represents the CMPT, CLS, and remaining tokens from the last encoder layer. Note that the pretrained encoder parameters $\Theta_{f,m}$ remain frozen while we only learn $\Delta_{f,m}$.

We aim to train the CMPTs such that they act as a proxy of the class token of the missing modality at the inference time. In particular, we want the CMPT $\hat{\mathcal{T}}_{\text{CMPT},m_1}$ of modality $m_1$ to approximate the class token $\hat{\mathcal{T}}_{\text{CLS},m_2}$ of modality $m_2$, and vice versa. To achieve this, we use an alignment loss $\mathcal{L}_{\text{align}}$, defined as

$$\mathcal{L}_{\text{align}} = \frac{1}{N_{\mathcal{D}_c}} \sum_{x \in \mathcal{D}_c} \begin{aligned} &\mathcal{L}_{\text{MSE}}(\hat{\mathcal{T}}_{\text{CMPT},m_1}, \hat{\mathcal{T}}_{\text{CLS},m_2}) + \\ &\mathcal{L}_{\text{MSE}}(\hat{\mathcal{T}}_{\text{CMPT},m_2}, \hat{\mathcal{T}}_{\text{CLS},m_1}), \end{aligned} \tag{4}$$

where $\mathcal{L}_{\text{MSE}}$ denotes the mean squared error loss and $N_{\mathcal{D}_c}$ denotes the number of samples in the training data where both modalities are present.

### 3.4 Gating module and fusion

After computing the final output tokens $\hat{\mathcal{T}}_m$, we extract the class token ($\hat{\mathcal{T}}_{\text{CLS},m}$) and the CMPT ($\hat{\mathcal{T}}_{\text{CMPT},m}$) for both modalities and pass them through the gating module. If both modalities are available, the gating module returns the class tokens from each modality. If one modality is missing, the gating module substitutes the missing class token with the CMPT from the available modality. The returned tokens are then fused using additive fusion to produce the final fused feature token $\mathcal{T} \in \mathbb{R}^{1\times d}$ as

$$\mathcal{T} = \begin{cases} \hat{\mathcal{T}}_{\text{CLS},m_1} + \hat{\mathcal{T}}_{\text{CLS},m_2}, & m_1, m_2 \text{ are available,} \\ \hat{\mathcal{T}}_{\text{CLS},m_2} + \hat{\mathcal{T}}_{\text{CMPT},m_2}, & m_1 \text{ is missing,} \\ \hat{\mathcal{T}}_{\text{CLS},m_1} + \hat{\mathcal{T}}_{\text{CMPT},m_1}, & m_2 \text{ is missing.} \end{cases} \tag{5}$$

The fused feature token $\mathcal{T}$ is then passed through a linear classifier head to make a prediction $\hat{y}$ as

$$\hat{y} = h(\mathcal{T}; \Theta_h). \tag{6}$$

$\Theta_h$ represents the parameters of the classifier head that are learned during training.

Finally, we learn the low-rank factors for each encoder ($\{\Delta_{f,m}\}$) and classifier head ($\Theta_h$) by minimizing the task-specific loss $\mathcal{L}_{\text{task}}$, which for our purposes is a standard cross-entropy loss for classification tasks, combined with the alignment loss:

$$\mathcal{L}_{\text{total}} = \mathcal{L}_{\text{task}} + \lambda \mathcal{L}_{\text{align}}, \tag{7}$$

where $\lambda$ is a hyperparameter that we set to $\lambda = 0.20$ in all our experiments. As discussed in Section 4.6.3, this value provides a balance in performance across different scenarios.

### 3.5 Handling missing modalities

Our model can handle missing modalities during both training and inference. During training on complete modality data, we apply random modality dropout, where we either use both modalities or randomly drop one in each iteration. This way, the model learns to make correct predictions in missing modality scenarios using the CMPTs from the available modality. When training data contains missing modalities, we learn the CMPTs by optimizing the alignment loss using only the samples with complete modalities, as shown in equation 4. For missing modalities, we employ placeholders (e.g., an empty string for text and zeros for images, audio, and video) following standard practices (Lee et al., 2023; Reza et al., 2024a), and use the CMPT from the available modality as an approximation to make predictions, as described in equation 5.

To enhance training efficiency, we leverage LoRA (Hu et al., 2022) with a rank of 1 to fine-tune the encoders in a parameter-efficient manner. LoRA layers are inserted after the query, key, value, and output layers in every attention blocks within the encoders. Our experiments demonstrate that LoRA with rank 1 is sufficient to achieve better performance compared to existing state-of-the-art methods, as discussed in Section 4.4 and Section A4. Our method is also compatible with newer parameter-efficient adaptation techniques such as DoRA (Liu et al., 2024) and VeRA (Kopiczko et al., 2024), as discussed in Section A10.

**Table 1:** Performance comparison when modality gets missing during both training and inference. We train and evaluate our model with same modality ratio following existing baselines (Lee et al., 2023; Jang et al., 2024; Kim & Kim, 2024) and report the mean and standard deviation (SD) of three independent runs with different seeds.

| Datasets | % availability | | ViLT | MAP (2023) | | MSP | VisualBERT | Ma et al. | Kim & Kim | MuAP (2024) | | CMPT (Ours) |
|---|---|---|---|---|---|---|---|---|---|---|---|---|
| | Image | Text | (2021) | Input | Attn. | (2024) | (2019) | (2022) | (2024) | Head | Cross | Mean ± SD |
| MM-IMDb (F1-Macro) | 30% | 100% | 37.61 | 46.30 | 44.74 | 47.45 | 38.63 | 46.63 | 56.03 | 47.21 | 46.73 | **60.21** ± **0.40** |
| | 65% | 65% | 36.30 | 42.66 | 41.56 | 42.03 | 37.23 | 41.28 | 49.24 | 42.57 | 43.92 | **54.04** ± **0.23** |
| | 100% | 30% | 34.71 | 39.22 | 38.16 | 38.34 | 36.41 | 38.65 | 43.21 | 41.37 | 39.88 | **52.61** ± **0.12** |
| UPMC Food-101 (Accuracy) | 30% | 100% | 76.93 | 86.18 | 86.05 | 86.34 | 77.41 | 86.38 | 87.11 | 86.90 | 86.59 | **87.48** ± **0.13** |
| | 65% | 65% | 69.03 | 79.08 | 78.09 | 78.89 | 71.06 | 78.58 | 82.67 | 77.87 | 78.95 | **82.83** ± **0.25** |
| | 100% | 30% | 66.29 | 74.53 | 72.57 | 73.77 | 67.78 | 73.41 | 78.81 | 74.61 | 74.60 | **80.37** ± **0.18** |

**Table 2:** Performance comparison when a complete modality gets missing during inference. We report % accuracy for all the datasets. Our method shows overall better performance compared to recent state-of-the-art methods.

| Datasets | Available Modality | Missing Modality | FiLM (2018) | BiGated (2018) | OGM-GE (2022) | QMF (2023b) | MLA (2024) | CMPT (Ours) |
|---|---|---|---|---|---|---|---|---|
| CREMA-D | Audio | Video | 53.89 | 51.49 | 53.76 | 59.41 | 59.27 | **67.20** |
| | Video | Audio | 18.67 | 17.34 | 28.09 | 39.11 | 64.91 | **76.21** |
| | Audio - Video | - | 60.07 | 59.21 | 68.14 | 63.71 | 79.70 | **88.84** |
| KS | Audio | Video | 48.67 | 49.96 | 48.87 | 51.57 | 54.67 | **68.27** |
| | Video | Audio | 23.15 | 23.77 | 29.73 | 32.19 | 51.03 | **85.77** |
| | Audio - Video | - | 63.33 | 63.72 | 65.74 | 65.78 | 71.35 | **91.21** |
| UPMC Food-101 | Image | Text | 4.68 | 14.20 | 22.35 | 45.74 | 69.60 | **75.66** |
| | Text | Image | 85.84 | 85.79 | 85.17 | 84.13 | **86.47** | 85.31 |
| | Image - Text | - | 87.21 | 88.87 | 87.54 | 92.87 | 93.33 | **94.47** |

# 4 Experiments and Results

## 4.1 Datasets

We evaluate our approach on five popular multimodal datasets across different tasks. A brief description of each dataset is provided here, with further details in Section A2 in the appendix.

**UPMC Food-101** (Wang et al., 2015) is a multimodal classification dataset consisting of 101 food classes. It contains 90,704 image-text pairs, divided into a training set of 67,988 pairs and a test set of 22,716 pairs.

**MM-IMDb** (Arevalo et al., 2017) has 25,959 samples with image and text modalities for multi-label movie genre classification. It is split into 15,552 training, 2,608 validation, and 7,799 test samples across 23 classes.

**Kinetics-Sound (KS)** (Arandjelovic & Zisserman, 2017) is a subset of Kinetics400 dataset (Kay et al., 2017) for human action recognition using audio and video as input modalities. It has 31 classes, with 14,739 samples in the training set and 2,594 samples in the test set.

**Audio-Visual Event (AVE)** dataset (Tian et al., 2018) contains 4,143 10-second videos for multimodal event localization. It is divided into train/val/test sets containing 3,339/402/402 samples across 28 categories.

**CREMA-D** dataset (Cao et al., 2014) includes 7,442 short video clips from 91 actors expressing six emotion categories. It contains 6,698 training and 744 test samples for multimodal emotion recognition.

## 4.2 Implementation details

**Vision-Language datasets.** We use pretrained ViT-B (Dosovitskiy et al., 2021) from CLIP model (Radford et al., 2021) as image encoder and BERT-base-uncased model (Lu et al., 2019) as text encoder. The max token length is set to 40 for UPMC Food-101 and 256 for MM-IMDb dataset, respectively.

**Table 3:** Performance comparison with both modalities available during training and inference on Image-Text datasets. We report % accuracy for UPMC Food-101 and F1-Macro/F1-Micro score for MM-IMDb dataset. Here, '-' indicate results that are not reported in corresponding papers.

| Methods | Parameters (M) Total | Learnable | UPMC Food-101 | MM-IMDb |
|---|---|---|---|---|
| ViLT (2021) | 112.63 | 112.63 | 92.00 | 55.30 / 64.70 |
| MBT (2021b) | 196.00 | 196.00 | 93.56 | 59.60 / 64.81 |
| MMBT (2019) | 170.00 | 170.00 | 94.10 | 60.80 / 66.10 |
| MMLoRA (2023) | 196.10 | 196.10 | 93.70 | 61.70 / 67.20 |
| PMF (2023) | 198.43 | 2.54 | 91.51 | 58.77 / 64.51 |
| PMF-L (2023) | 643.95 | 4.44 | 91.68 | 61.66 / 66.72 |
| MLA (2024) | 218.26 | 218.26 | 93.33 | - / - |
| **CMPT (Ours)** | 195.51 | **0.27** | **94.47** | **63.58 / 69.23** |

**Table 4:** Performance (% accuracy) comparison with both modalities available during training and inference on Audio-Video datasets. Here, '*' denotes results generated using existing code.

| Method | Parameters (M) Total | Learnable | KS | AVE | CREMA-D |
|---|---|---|---|---|---|
| G-Blend (2020a) | 22.36 | 22.36 | 62.20 | 65.50 | 58.70 |
| FiLM (2018) | 22.35 | 22.35 | 63.33 | - | 60.07 |
| BiGated (2018) | 22.35 | 22.35 | 63.72 | - | 59.21 |
| OGM-GE (2022) | 22.35 | 22.35 | 65.74 | 76.90 | 68.14 |
| QMF (2023b) | 22.36 | 22.36 | 65.78 | 68.16* | 63.71 |
| PMR (2023b) | 22.36 | 22.36 | - | 66.40 | 61.80 |
| MLA (2024) | 22.37 | 22.37 | 71.35 | 64.68* | 79.70 |
| MBT (2019) | 172.00 | 172.00 | 85.00 | 77.80 | - |
| MMLoRA (2023) | 172.19 | 172.19 | **91.40** | 96.20 | 88.60 |
| **CMPT (Ours)** | 172.16 | **0.17** | 91.21 | **96.77** | **88.84** |

**Audio-Video datasets.** We use AST model (Gong et al., 2021) pretrained on AudioSet dataset (Gemmeke et al., 2017) as audio encoder and ViT-B (Dosovitskiy et al., 2021) from CLIP model (Radford et al., 2021) as video encoder. Raw audio is converted to a 128-bin mel spectrogram at 16 kHz, with a maximum length of 1024 frames. For video, we sample 3 frames randomly while training and uniformly while testing. Other pre-processing steps are similar to (Zhang et al., 2024; Peng et al., 2022; Du et al., 2023).

**Hyperparameter settings.** We use Python 3.8.19 and PyTorch 2.2.2 for training and evaluating our models. All the models are trained using two NVIDIA RTX 2080Ti GPUs. For vision-language datasets, we set the learning rate to $10^{-3}$ and train the models for 10 epochs with a batch size of 8. For audio-video datasets, the learning rate is set to $5 \times 10^{-5}$ and models are trained for 100 epochs with a batch size of 4. We use AdamW (Loshchilov & Hutter, 2019) optimizer with $\epsilon = 10^{-8}$ and weight decay = 0.02. While training, we utilize cross entropy loss, polynomial learning rate scheduler with power=0.9 and set the first 5 epoch as warm-up. The learning rate is set to 0.1 times the original learning rate during warm-up epochs. We set LoRA (Hu et al., 2022) rank = 1 and insert them after query, key, value and output layers of each transformer block. Further details with all the hyperparameters can be found in Section A3 in the appendix.

For each task and dataset, we compare against state-of-the-art published work. For this reason, the set of baselines varies across experiments based on the availability of published results. In each table, the best and second best results are bold and underlined, respectively.

## 4.3 Performance when modalities are missing

Existing literature considers missing modalities at both training and inference or only at inference. Our method achieves state-of-the-art performance in both scenarios, as detailed in the next two sections.

### 4.3.1 Missing modality during training and inference

In the first setup, a modality is missing at a fixed rate during both training and inference, as studied by Lee et al. (2023); Jang et al. (2024); Kim & Kim (2024). We follow their experiment setup and compare our method against seven state-of-the-art approaches on MM-IMDb and UPMC Food-101 datasets. As shown in Table 1, our method consistently outperforms all baselines across various missing modality configurations. Specifically, compared to the most recent approach by Kim & Kim (2024), we achieve a 4.18%–9.40% improvement in F1-macro score on MM-IMDb dataset and a 0.16%–1.56% improvement in accuracy on UPMC Food-101 dataset across different missing modality scenarios. It is worth noting that while the performance of Kim & Kim (2024) on UPMC Food-101 dataset is comparable to ours in two scenarios, our method still performs best in every configuration. These results also suggest that the CMPTs can provide good proxy for the missing class tokens, which significantly improves the model robustness. To ensure a reliable evaluation, we run our method with three different random seeds and report the mean and standard deviation (SD) for various missing scenarios. The low SD values indicate the stability and reliability of our

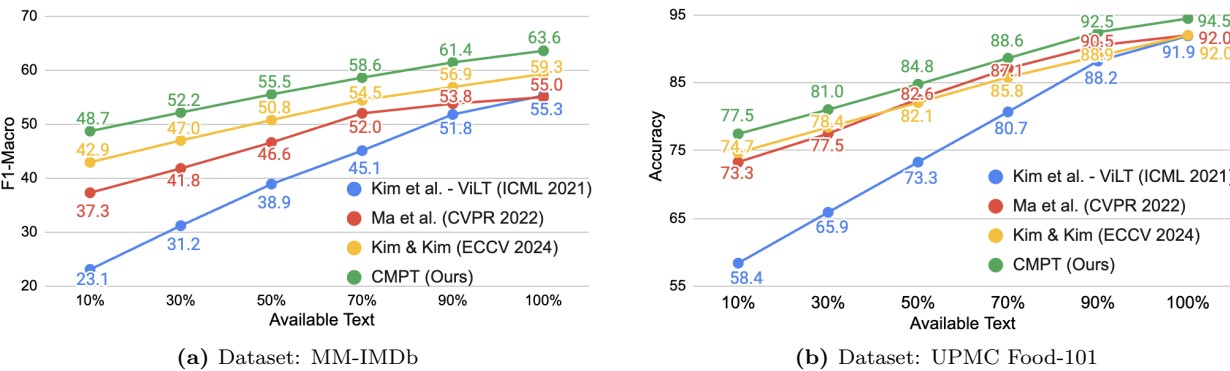

**Figure 2:** Generalization to varying missing rates during inference. Models are trained with 100% image + 100% text and evaluated with 100% image + $x$% text following Ma et al. (2022); Kim & Kim (2024). Our approach demonstrates better generalization, particularly under severe modality loss.

method. We also assessed the performance of the models with different distributions of modality ratios at training and testing, for which the results are presented in Section A5 of the appendix.

### 4.3.2 Missing modality during inference only

In the second setup, following the work of Zhang et al. (2024), we evaluate our model when all modalities are available during training, but one modality is completely missing during inference. As shown in Table 2, our method outperforms existing baselines in most cases. For instance, when the audio modality is missing, all methods except MLA (Zhang et al., 2024) experience a significant performance drop on the CREMA-D and Kinetics-Sound (KS) datasets. On the CREMA-D dataset, our method achieves 7.93% and 11.30% improvements over the most recent MLA approach, when video and audio are missing, respectively. Similarly, on the KS dataset, our method outperforms MLA with 13.60% and 34.74% improvements when video and audio are missing, respectively. While MLA performs slightly better when the image modality is missing on the UPMC Food-101 dataset, it suffers a larger performance drop when text is missing. In contrast, our method maintains a balanced performance across different missing modalities. Furthermore, across all datasets, our method achieves the best performance when all modalities are available and demonstrates superior robustness compared to other approaches. These results highlight the effectiveness of our method in handling scenarios where a modality is entirely absent during inference.

### 4.4 Performance when all modalities are available

A method designed to handle missing modalities should not compromise performance when all modalities are present. In this subsection, we demonstrate that our approach also achieves comparable or superior performance to state-of-the-art models across five datasets when all modalities are available.

To evaluate this, we train and test our method when all modalities are available during training and inference. Results in Table 3 demonstrate that our proposed method achieves state-of-the-art performance on both UPMC Food-101 and MM-IMDb datasets. Our method surpasses the most recent MLA approach (Zhang et al., 2024) by 1.14% on UPMC Food-101. Our method achieves 1.88% and 2.03% improvement in F1-macro and F1-micro scores, respectively, over second-best performing MMLoRA (Du et al., 2023) on MM-IMDb.

We present the results for audio-video datasets in Table 4, where we compare our method against nine state-of-the-art methods. Our approach consistently achieves comparable or superior performance across all three datasets, while using the fewest number of learnable parameters. These results highlights that incorporating CMPTs does not introduce any drawback when all modalities remain available.

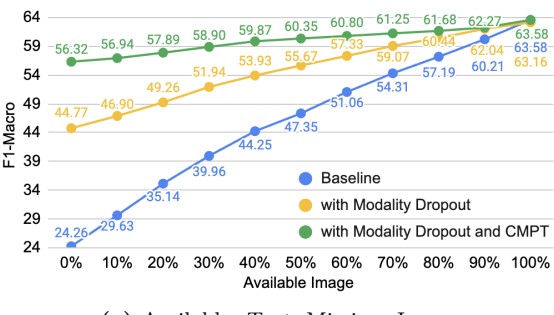 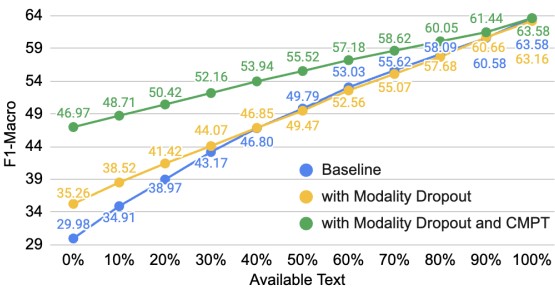

**(a)** Available: Text, Missing: Image.     **(b)** Available: Image, Missing: Text.

**Figure 3:** Effectiveness of CMPTs on the MM-IMDb dataset. All models are trained with 100% image and 100% text data, then evaluated under varying amount of missing modalities. CMPTs achieve significant performance improvement, especially under severe missing modality scenarios.

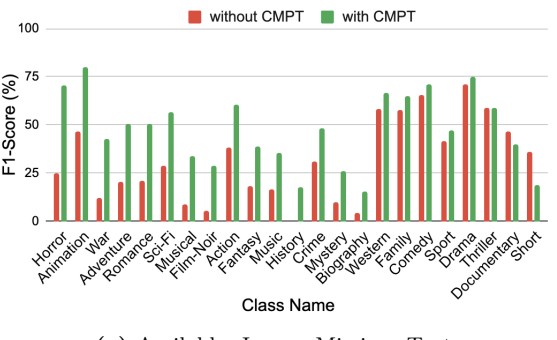 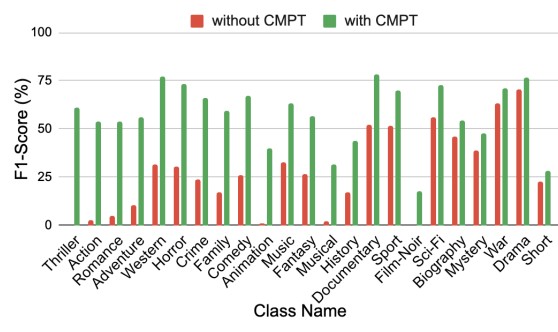

**(a)** Available: Image, Missing: Text.     **(b)** Available: Text, Missing: Image.

**Figure 4:** CMPTs improve performance across most of the classes when modalities are missing. However, it fails to accurately predict certain classes (e.g., short/documentary) in the absence of text, likely due to the image modality's insufficient information for those classes. Classes are sorted based on the amount of performance improvement.

## 4.5 Generalization to different missing rates

In real-world scenarios, the missing rates for modalities during inference can be unpredictable. Any method designed to handle missing modalities needs to generalize effectively across varying levels of missing modality rates during inference. To evaluate this capability, we trained our models on complete multimodal data and assessed their performance under varying levels of missing modalities during inference. Specifically, we follow the experiment setup similar to Ma et al. (2022); Kim & Kim (2024) where models are trained with 100% image + 100% text data and evaluated with 100% image + $x$% text.

As illustrated in Figure 2, our method consistently outperforms all baselines across different missing scenarios, demonstrating superior robustness. We observe a decline in performance of all methods as the amount of available text decreases, but our approach maintains stronger resilience compared to other methods. For instance, on the MM-IMDb dataset, our method achieves a 5.8% higher F1-Macro score than the most recent method by Kim & Kim (2024) and a 25.6% improvement over ViLT (Kim et al., 2021) when only 10% of text is available. A similar trend is observed on the UPMC Food-101 dataset, where our method consistently yields superior performance, confirming its robustness to missing modalities at inference.

Note that published work by Ma et al. (2022); Kim & Kim (2024); Kim et al. (2021) did not report results for different missing rates of images, which prevents us from reporting a direct comparison in that setting. Nevertheless, our results highlight the effectiveness of CMPTs in handling different missing modality scenarios, making it a more generalizable solution for real-world applications.

**Table 5:** Ablation study on the alignment loss weight $\lambda$. We report performance on the UPMC Food-101 and AVE datasets across both complete and missing modality scenarios.

| Dataset | Available Modality | Missing Modality | $\lambda = 0.0$ | $\lambda = 0.1$ | $\lambda = 0.2$ | $\lambda = 0.3$ | $\lambda = 0.4$ | $\lambda = 0.5$ |
|---|---|---|---|---|---|---|---|---|
| UPMC Food-101 | Image | Text | 73.12 | 73.48 | **75.66** | 73.72 | 74.39 | 74.46 |
| | Text | Image | 84.31 | 84.79 | **85.31** | 84.51 | 84.81 | 84.88 |
| | Image - Text | - | 93.96 | 94.14 | **94.47** | 94.13 | 94.34 | 94.21 |
| AVE | Audio | Video | 84.82 | 84.33 | **85.21** | 84.82 | 84.33 | 84.82 |
| | Video | Audio | 82.83 | 83.08 | 84.58 | **85.32** | 81.84 | 84.83 |
| | Audio - Video | - | 95.77 | 96.02 | **96.77** | 96.52 | 96.52 | 96.27 |

## 4.6 Ablation studies

### 4.6.1 Effectiveness of CMPTs

To evaluate the effectiveness of CMPTs in handling missing modalities, we compare our method against two baselines: (i) a standard model trained without modality dropout or CMPTs (baseline) and (ii) a model trained with modality dropout, where modalities are randomly removed during training to improve robustness to missing data. For a fair comparison, all models were fine-tuned on the downstream task using LoRA with a rank of 1.

Figure 3 presents the main results for MM-IMDb dataset, where the baseline model experiences a sharp performance drop as the proportion of missing modalities increases. While modality dropout improves robustness, the performance gap remains large. Incorporating CMPTs further enhance performance, achieving a 32.06% improvement over the baseline and 11.55% over modality dropout when the image modality is entirely missing as shown in Figure 3a. A similar trend is observed when text is missing as shown in Figure 3b. Notably, CMPTs become increasingly effective as the amount of missing data increases, demonstrating its strong ability to substantially mitigate performance degradation in the presence of severe modality loss. Similar trend appears in UPMC Food-101 dataset, which we discuss in Sec. A6 in the appendix.

### 4.6.2 Effects of CMPTs on different classes

When analyzing the effectiveness of CMPTs, a key question is whether the overall performance gains are driven by just a few classes or if CMPTs consistently improves performance across most classes. To address this, we conducted a per-class performance analysis with and without CMPTs.

As shown in Figure 4, CMPTs improve performance across nearly all classes in both text-missing and image-missing scenarios. Notably, it significantly mitigates performance degradation in modality-sensitive classes. For example, when text is missing, classes like *Horror*, *Animation*, and *War* experience severe drops in performance, but CMPTs effectively recover much of the loss (Figure 4a). Similarly, when images are missing, CMPTs provide substantial gains in *Thriller*, *Action*, and *Romance* (Figure 4b). A slight drop is observed in *Short* and *Documentary* classes when text is missing, likely due to insufficient information in the image modality alone for those classes. These results demonstrate that CMPTs enhance performance across most classes, leading to a strong overall performance improvement. A similar trend is observed on the UPMC Food-101 dataset which is discussed in Section A7 of the appendix.

### 4.6.3 Determining the Optimal Alignment Loss Weight $\lambda$

To determine the optimal value for the alignment loss weight $\lambda$, we performed an ablation study on the UPMC Food-101 and AVE datasets, varying $\lambda$ from 0.0 to 0.5. The results, shown in Table 5, indicate that the choice of $\lambda$ is crucial for performance in different scenarios. For our experiments, $\lambda = 0.20$ provides an overall better performance compared to other values. On the UPMC Food-101 dataset, setting $\lambda = 0.20$ consistently achieves the best performance across all conditions: with missing text, missing image, and with both modalities available. Using $\lambda = 0.0$, which is equivalent to removing the alignment loss, results in a significant performance drop, highlighting the importance of the alignment objective.

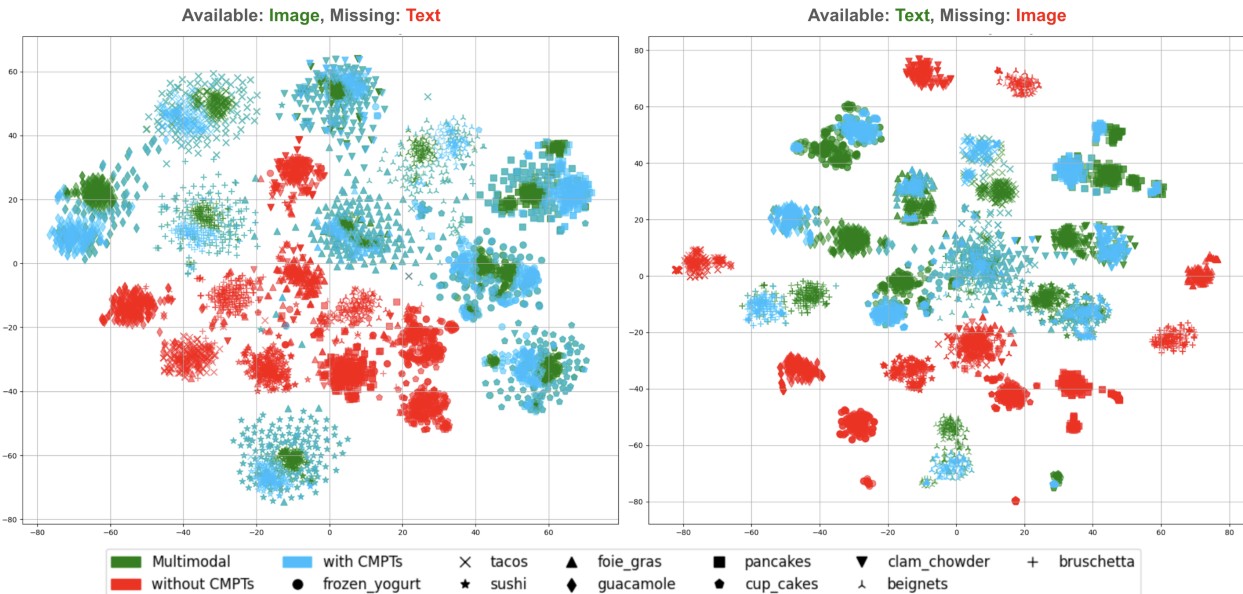

**Figure 5:** t-SNE plots of the fused feature tokens $\mathcal{T}$ for ten classes under different modality settings. The t-SNE plots show that fused features without CMPTs (red) deviate significantly from the complete multimodal features (green). Incorporating CMPTs (blue) provides embeddings that closely align with the complete multimodal features, indicating improved semantic alignment and robustness.

On the AVE dataset, the results show a similar trend. A weight of $\lambda = 0.20$ yields the best performance when video is missing and when both modalities are available. $\lambda = 0.30$ shows a slight performance improvement in the video-only scenario. Given that $\lambda = 0.20$ demonstrates superior or near-optimal performance consistently across both datasets, we set $\lambda = 0.20$ for all our experiments, as it provides a balance in performance across different modality conditions.

### 4.6.4 Qualitative Analysis of CMPT Effectiveness

To gain deeper insights into how CMPTs enhance robustness in missing modality scenarios, we conducted a qualitative analysis on the UPMC Food-101 dataset using both fused feature projections and attention visualizations. Figure 5 presents t-SNE plots of the fused feature $\mathcal{T}$ for ten representative food classes where CMPTs yield notable performance gains. We compare three settings: (i) complete multimodal input (green), (ii) missing modality without CMPTs (red), and (iii) missing modality with CMPTs (blue). The t-SNE plots show that fused features without CMPTs (red) deviate significantly from the complete multimodal features (green) in missing modality scenarios. Incorporating CMPTs (blue) provides embeddings that closely align with the complete multimodal features, indicating improved semantic alignment and robustness.

We also analyze the attention maps from the last layer of the vision encoder for the CLS token and CMPTs in the text-missing scenario, as shown in Figure 6. We selected random examples from six classes where fused features without CMPTs provide incorrect predictions, but including the CMPTs leads to correct predictions (approximately 15% samples in the dataset show this behavior). The attention maps show that CLS tokens attend to semantically-irrelevant regions, which likely leads to incorrect predictions. In contrast, CMPTs attend to semantically-relevant regions for the respective classes and lead to accurate predictions. These visualizations demonstrate that CMPTs not only act as proxies for the missing modality but also reshape the model's attention to reflect modality-relevant cues. For the sake of completeness, we have also included attention maps for some cases when the fused features with and without CMPTs provide correct predictions (approximately 60% samples show this pattern) and when the fused features with CMPTs lead to incorrect predictions (approximately 3% samples show this pattern) in Section A8.

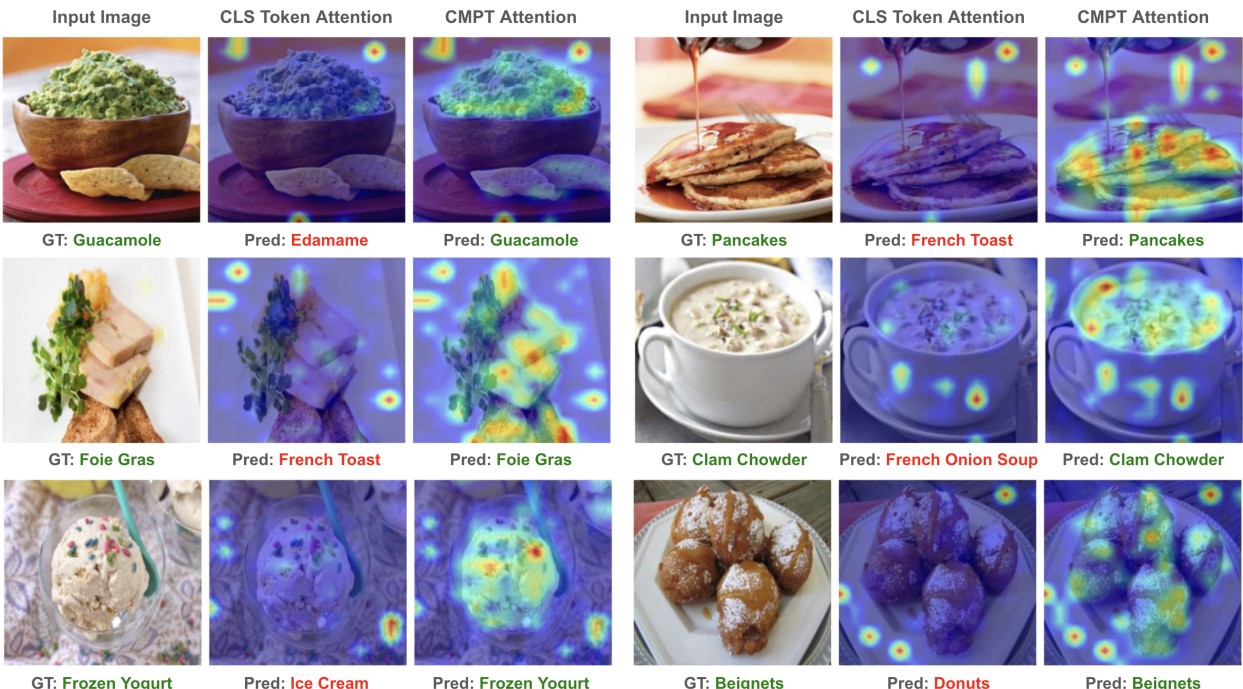

**Figure 6:** Attention map visualizations from the last layer of the vision encoder for the CLS token and CMPTs in the text-missing scenario on the UPMC Food-101 dataset. The attention maps show that CLS tokens attend to semantically-irrelevant regions, which likely leads to incorrect predictions. In contrast, CMPTs attend to semantically-relevant regions for the respective classes and lead to accurate predictions.

## 5 Limitations and Future Directions

While our approach demonstrates consistent performance improvements over existing methods in different missing modality scenarios, the remaining performance drop—especially in extreme cases—indicates room for further improvement. Additionally, we observe that certain classes are heavily dependent on a specific modality, which leads to significant performance drop in those classes when that specific modality is missing. A natural question arises as how can we design a better training strategy that encourages more uniform reliance on different modalities while maintaining overall performance in both complete and missing modality scenarios? Future work can also explore developing a general framework for explaining modality contribution and expected performance drop under different missing modality scenarios. Finally, this paper primarily focused on two-modality settings due to the availability of standard benchmarks and baselines that enable direct and fair comparisons. Extending the CMPT framework is a natural next step, and we are actively investigating this direction as part of our ongoing work.

## 6 Conclusion

We presented a simple yet effective framework for utilizing pre-trained unimodal encoders for robust multimodal learning. Our proposed framework enables training of multimodal systems that offer robust performance under different scenarios of missing modalities during training and inference. To compensate for missing modalities, we propose to learn cross-modal proxy tokens (CMPTs) to approximate the missing modality class tokens from the available modality using an alignment loss during training. Instead of training the unimodal encoders from scratch, we only learn rank-1 adapters along with task-specific classifier heads. Our extensive evaluations across diverse tasks and datasets demonstrate the robustness and adaptability of our approach in both complete and missing modality scenarios. Our analysis also reveals that CMPTs can generalize well to different missing scenarios and enhance performance across most of the classes to provide

an overall performance boost. Overall, our approach provides a robust and efficient solution for multimodal learning that maintains high performance in different real-world scenarios.

## Acknowledgment

This work is supported in part by NSF awards 2046293 and 2406199 and Amazon Gift awards.

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

# A Appendix

## A1 Related Work on Multimodal Learning

In this section, we provide a general overview of multimodal learning, which involves integrating information from different input sources to improve performance on downstream tasks. We can broadly divide multimodal learning approaches into two main categories: designing specialized models and adapting existing models.

**Designing specialized models** to effectively fuse multiple input sources is a common strategy in multimodal learning. For vision-language tasks, models such as ViLT (Kim et al., 2021), ViLBERT (Lu et al., 2019), and Visual-BERT (Li et al., 2019) process image patches and text tokens directly within a transformer architecture for tasks like image captioning and visual question answering. MMBT (Kiela et al., 2019) integrates both text and images within a BERT-based framework (Devlin et al., 2019) to model cross-modal dependencies. MBT (Nagrani et al., 2021b) leverages fusion bottlenecks for efficient modality-specific interaction, particularly in audio-video tasks. Specialized multimodal models have also been proposed for other applications, including action recognition (Woo et al., 2023; Zhou et al., 2022), segmentation (Zhang et al., 2023a; Wang et al., 2022b; Reza et al., 2024b), and sentiment analysis (Woo et al., 2023; Tsai et al., 2019). While these models excel in specific tasks, extending them to new modalities or domains often proves to be challenging and resource-intensive, as it requires retraining or significant modification of the model architecture.

**Adapting existing models** offers an alternative approach to multimodal learning, focusing on the use of pre-trained unimodal models and adapting them for multimodal tasks. One such strategy is gradient modulation, where techniques like G-Blend (Wang et al., 2020a) modulate gradients to combine unimodal and multimodal knowledge. OGM-GE (Peng et al., 2022) improves upon this by emphasizing modality-specific information, enhancing the model's ability to leverage unique features from each input source. PMR (Fan et al., 2023b) further refines this approach by modulating parameter groups based on the relevance of each modality, while QMF (Zhang et al., 2023b) introduces quadratic modulation for adaptive fusion. More recently, MLA (Zhang et al., 2024) proposed alternating adaptation, which toggles between unimodal adaptation and gradient modulation to enable more efficient cross-modal learning. Another notable approach, MMLoRA (Du et al., 2023), uses a two-phase strategy: unimodal fine-tuning followed by multimodal adaptation, which helps achieve optimal performance across a variety of tasks. However, while these methods allow for effective cross-modal learning, they often require significant computational resources and learning a large number of parameters to achieve optimal performance on downstream tasks.

## A2 Detailed Dataset Description

**UPMC Food-101** dataset (Wang et al., 2015) is a popular and challenging multimodal classification dataset. It has two input modalities: image and text. The dataset has 90,704 samples divided into training and test sets having 67,988 and 22,716 samples, respectively. It contains 101 classes, which are the same as those in the ETHZ Food-101 dataset (Bossard et al., 2014). The samples are noisy, each category contains around 5% irrelevant images as they were collected in an uncontrolled environment without any human human intervention during the data collection process.

**MM-IMDb** dataset (Arevalo et al., 2017) is a widely used multimodal dataset for multi-label movie genre classification task. It consists of 25,959 samples, each containing both image and text as input modality. The dataset is divided into three subsets: 15,552 samples for training, 2,608 for validation, and 7,799 for testing. It has 23 classes and each sample can have one or more genre. This dataset serves as a standard benchmark for evaluating multimodal models in the multi-label classification task.

**Kinetics-Sound (KS)** dataset (Arandjelovic & Zisserman, 2017) is a subset of the Kinetics-400 dataset (Kay et al., 2017). It is used for human action recognition using audio and video as input modalities. This dataset includes 31 different action classes that span a variety of everyday human activities, making it suitable for evaluating multimodal models. The dataset is split into two subsets: a training set containing 14,739 samples and a test set containing 2,594 samples.

**Table 6:** List of all the hyperparameters used in our experiments.

| Hyperparameters | Image-Text Datasets | Audio-Video Datasets |
|---|---|---|
| Learning Rate | $10^{-3}$ | $5 \times 10^{-5}$ |
| Scheduler | Polynomial | Polynomial |
| Power | 0.9 | 0.9 |
| Num Epochs | 10 | 100 |
| Warmup Epochs | 5 | 5 |
| Optimizer | AdamW | AdamW |
| $\epsilon$ | $10^{-8}$ | $10^{-8}$ |
| Weight Decay | 0.02 | 0.02 |
| $\lambda$ | 0.20 | 0.20 |
| LoRA Rank | 1 | 1 |
| LoRA Alpha | 1 | 1 |
| LoRA Dropout | 0.1 | 0.1 |
| Image Encoder | ViT-B | ViT-B |
| Text Encoder | BERT-Base | - |
| Audio Encoder | - | AST |
| Image Resolution | $224 \times 224$ | $224 \times 224$ |
| Patch Size | $16 \times 16$ | $16 \times 16$ |
| Batch Size | 8 | 4 |
| Audio Sampling Rate | - | 16 kHz |
| Frequency Bins | - | 128 |
| Maximum Length | - | 1024 Frames |
| Frames Per Video | - | 3 |
| Python Version | 3.8.19 | 3.8.19 |
| PyTorch Version | 2.2.2 | 2.2.2 |
| GPU Model | RTX 2080Ti | RTX 2080Ti |
| Num GPUs | 2 | 1 |

**Audio-Visual Event (AVE)** Localization dataset (Tian et al., 2018) is a benchmark dataset used for multimodal event localization task. It contains 4,143 10-second videos. The dataset is divided into training, validation, and test sets, with 3,339 videos for training, 402 for validation, and 402 for testing. The dataset encompasses 28 different event categories providing a broad range of scenarios where both audio and visual signals are crucial for accurate event detection. All videos in the AVE dataset are collected from YouTube.

**CREMA-D** dataset (Cao et al., 2014) is used for multimodal emotion recognition. It has audio and video as input modalities. The dataset contains 7,442 short video clips performed by 91 actors. The clips cover six emotions: angry, happy, sad, neutral, disgust, and fear. Emotion labels were obtained from 2,443 crowd-sourced raters to ensure diverse evaluations. The dataset is divided into 6,698 samples for training and 744 samples for testing.

## A3    Implementation Details

In Section 4.2 of the main paper, we briefly discussed the hyperparameter settings used in our experiments. Here, we provide a complete list of all the hyperparameters and their values in Table 6. The "Image-Text Datasets" column includes the hyperparameters for UPMC Food-101 and MM-IMDb datasets, while the "Audio-Video Datasets" column lists the hyperparameters for Kinetics-Sound, AVE, and CREMA-D datasets.

**Table 7:** Performance comparison with different LoRA ranks ($r$). The models are trained on complete multimodal data and tested on both complete and missing modality scenarios. The best Accuracy (Acc.) for each scenario is highlighted in bold. Params (M) indicates the number of learnable parameters in millions.

| Datasets | Available Modality | Missing Modality | r = 1 | | r = 2 | | r = 4 | |
|---|---|---|---|---|---|---|---|---|
| | | | Params (M) | Acc. (%) | Params (M) | Acc. (%) | Params (M) | Acc. (%) |
| UPMC Food-101 | Image | Text | 0.271 | **75.66** | 0.376 | 75.62 | 0.671 | 70.38 |
| | Text | Image | | 85.31 | | **85.61** | | 83.92 |
| | Image - Text | - | | 94.47 | | **94.64** | | 93.30 |
| AVE | Audio | Video | 0.173 | **85.21** | 0.319 | 85.07 | 0.614 | 84.33 |
| | Video | Audio | | **84.58** | | 84.30 | | 80.84 |
| | Audio - Video | - | | **96.77** | | **96.77** | | 96.27 |
| CREMA-D | Audio | Video | 0.157 | **67.20** | 0.303 | 65.32 | 0.598 | 65.46 |
| | Video | Audio | | 76.21 | | 75.54 | | **76.61** |
| | Audio - Video | - | | **88.84** | | 87.63 | | 87.77 |
| KS | Audio | Video | 0.176 | 68.27 | 0.322 | **69.93** | 0.617 | 69.54 |
| | Video | Audio | | 85.77 | | 85.20 | | **86.43** |
| | Audio - Video | - | | **91.21** | | 91.06 | | 90.59 |

## A4    Determining the Optimal LoRA Rank

To determine the optimal LoRA rank, we conducted an ablation study evaluating ranks $r \in \{1, 2, 4\}$. For this analysis, we trained our models on modality-complete data and evaluated their performance in both complete and missing modality scenarios, with the results presented in Table 7.

Our analysis indicates that increasing the rank, and thereby the number of learnable parameters, does not guarantee a consistent improvement in performance. For instance, on the AVE dataset, a rank of $r = 1$ consistently achieves the highest accuracy. On the UPMC Food-101 dataset, performance between $r = 1$ and $r = 2$ is comparable. While $r = 2$ offers a marginal improvement in two scenarios, it requires a 38.7% increase in learnable parameters. In contrast, using $r = 4$ consistently resulted in degraded performance in both of these datasets compared to $r = 1$. A similar trend is observed for the CREMA-D dataset, where $r = 1$ performs best in two of the three scenarios. Conversely, the KS dataset presented mixed results, with different ranks yielding optimal performance in different scenarios.

Overall, a rank of $r = 1$ demonstrates competitive, and often superior performance across a majority of the tested scenarios. The performance degradation at higher ranks, such as $r = 4$, suggest that a larger number of learnable parameters may not be necessary for these tasks and could introduce a risk of overfitting. This aligns with the common characteristic of LoRA, where a low rank is often sufficient to capture essential task-specific information. Based on these observations, we use LoRA with $r = 1$ for all our experiments as it provides an optimal balance, delivering robust performance with the greatest parameter efficiency.

## A5    Different Modality Ratios During Training and Inference

A robust multimodal learning method should maintain strong performance even when the missing modality ratio during inference differs from that during training. To evaluate the robustness of our approach, we follow the experimental setup of Jang et al. (2024); Kim & Kim (2024) and train and test our model with varying modality ratios. Let $\eta$ denote the missing rate. We consider three scenarios: missing-text, missing-image, and missing-both. In the first two cases, $\eta\%$ of the samples contain only images or only text, while $(1-\eta)\%$ retain both modalities. For the missing-both case, $\frac{\eta}{2}\%$ of the samples contain only texts, $\frac{\eta}{2}\%$ contain only images, and $(1-\eta)\%$ include both modalities. We set $\eta = 70$ throughout our experiments, following existing baselines (Jang et al., 2024; Kim & Kim, 2024).

As summarized in Table 8, our method consistently outperforms existing state-of-the-art approaches in most scenarios. While the method proposed by Kim & Kim (2024) performs well in certain cases, our approach demonstrates superior performance across both the MM-IMDb and UPMC Food-101 datasets. To ensure a rigorous evaluation, we report the mean and standard deviation (SD) over three independent runs with

**Table 8:** Performance comparison when modality gets missing during both training and inference. We train and evaluate our model with different modality ratio following existing baselines (Jang et al., 2024; Kim & Kim, 2024) and report the mean and standard deviation (SD) of three independent runs with different seeds.

| Datasets | Missing Rate ($\eta$) | Train Image | Train Text | Inference Image | Inference Text | ViLT (2021) | MAP (2023) | MSP (2024) | Kim & Kim (2024) | CMPT (Ours) (Mean ± SD) |
|---|---|---|---|---|---|---|---|---|---|---|
| MM-IMDb (F1-Macro) | 70% | 30% | 100% | 30% | 100% | 37.63 | 47.18 | 47.45 | 56.03 | **60.21** ± **0.40** |
| | | | | 65% | 65% | 34.47 | 37.25 | 42.17 | 49.81 | **50.05** ± **0.53** |
| | | | | 100% | 30% | 30.00 | 24.31 | 34.32 | **43.07** | 36.10 ± 0.19 |
| | | 65% | 65% | 30% | 100% | 36.48 | 42.72 | 44.81 | 55.26 | **57.45** ± **0.21** |
| | | | | 65% | 65% | 36.54 | 40.84 | 42.03 | 49.24 | **54.04** ± **0.23** |
| | | | | 100% | 30% | 33.76 | 37.80 | 37.30 | 42.46 | **50.05** ± **0.29** |
| | | 100% | 30% | 30% | 100% | 27.25 | 20.17 | 34.11 | **54.67** | 44.72 ± 0.46 |
| | | | | 65% | 65% | 32.51 | 29.79 | 36.97 | 49.09 | **49.26** ± **0.42** |
| | | | | 100% | 30% | 34.26 | 36.89 | 38.34 | 43.21 | **52.61** ± **0.12** |
| UPMC Food-101 (Accuracy) | 70% | 30% | 100% | 30% | 100% | 76.40 | 86.34 | 86.34 | 87.11 | **87.48** ± **0.13** |
| | | | | 65% | 65% | 59.03 | 57.92 | 71.88 | **81.22** | 77.52 ± 0.29 |
| | | | | 100% | 30% | 41.94 | 29.71 | 56.87 | **75.41** | 67.60 ± 0.26 |
| | | 65% | 65% | 30% | 100% | 73.62 | 85.55 | 85.91 | 87.35 | **87.36** ± **0.20** |
| | | | | 65% | 65% | 68.60 | 78.49 | 78.89 | 82.67 | **82.83** ± **0.25** |
| | | | | 100% | 30% | 64.25 | 71.17 | 71.58 | 78.13 | **78.40** ± **0.09** |
| | | 100% | 30% | 30% | 100% | 43.67 | 27.46 | 57.22 | **86.90** | 85.34 ± 0.15 |
| | | | | 65% | 65% | 54.75 | 50.79 | 65.13 | 82.35 | **82.78** ± **0.22** |
| | | | | 100% | 30% | 66.01 | 73.71 | 73.77 | 78.81 | **80.37** ± **0.18** |

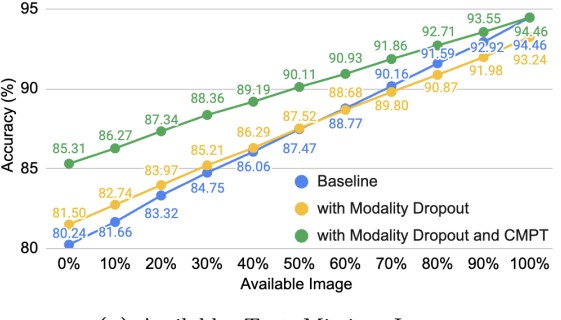

**(a)** Available: Text, Missing: Image.

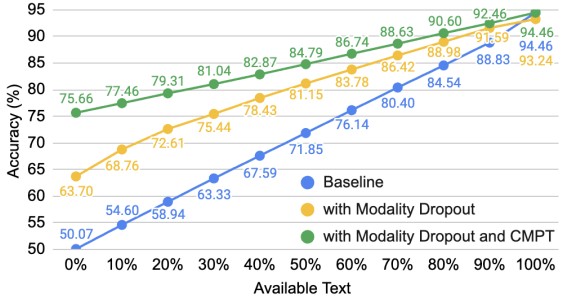

**(b)** Available: Image, Missing: Text.

**Figure 7:** Effectiveness of CMPTs in handling missing modalities on UPMC Food-101 dataset. CMPTs significantly improves performance when modalities are severely missing. Models are trained on 100% image + 100% text and tested with varying levels of missing modalities.

different seeds. The low SD values further indicate the stability and reliability of our method. These results substantiate our claim that CMPTs can effectively estimate missing modality features, enhancing robustness to varying missing modality conditions. Other methods reported in Table 1 do not report results for this specific experimental setup, and thus we are unable to compare with them.

## A6 Effectiveness of CMPTs

We extend the study from Section 4.6.1 to evaluate the effectiveness of CMPTs in handling missing modalities on the UPMC Food-101 dataset. We compare it against two baselines: (i) a standard model trained without modality dropout or CMPTs (baseline) and (ii) a model trained with modality dropout, where modalities

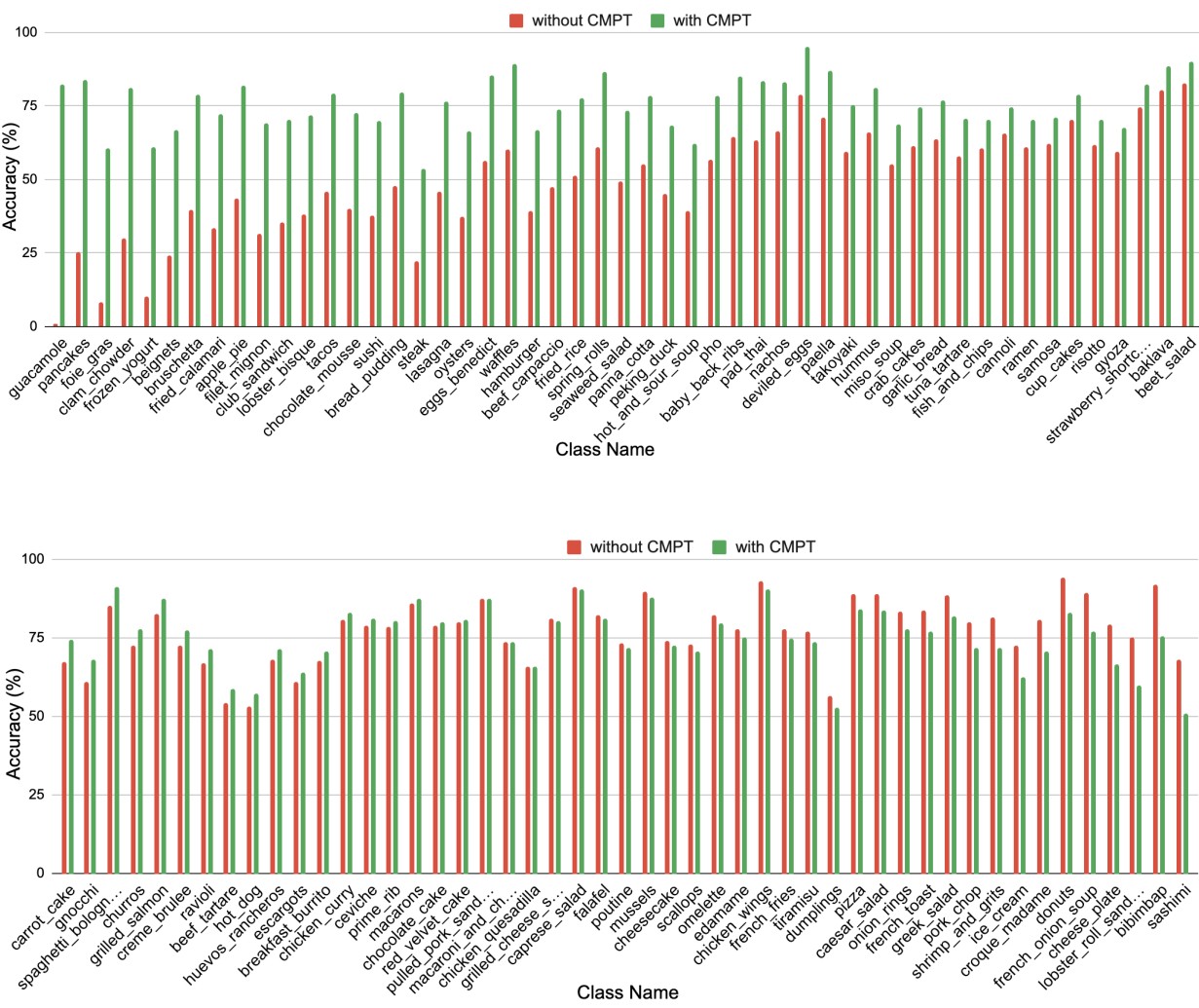

**Figure 8:** CMPTs improve performance across most of the classes when *text* modality is missing on UPMC Food-101 dataset. However, it fails to accurately predict certain classes (e.g., bibimbap/sashimi), likely due to the image modality's insufficient information for those classes. Classes are sorted based on the amount of performance improvement.

are randomly removed during training to improve robustness to missing data. For a fair comparison, all models were fine-tuned on the downstream task using LoRA with a rank of 1.

As illustrated in Figure 7, the baseline model experiences a significant performance drop as the proportion of missing modalities increases. While modality dropout improves robustness, it fails to fully recover the lost performance. Incorporating CMPTs further enhances performance, achieving a 5.07% improvement over the baseline and 3.81% over modality dropout when the image modality is entirely missing as shown in Figure 7a. A similar trend is observed when text is missing as shown in Figure 7b, where CMPTs achieves 25.59% and 11.96% higher accuracy compared to the baseline and modality dropout, respectively. Consistent with our previous findings, CMPTs becomes increasingly effective as the proportion of missing data increases, demonstrating its strong capability to substantially mitigate performance degradation in the presence of severe modality loss.

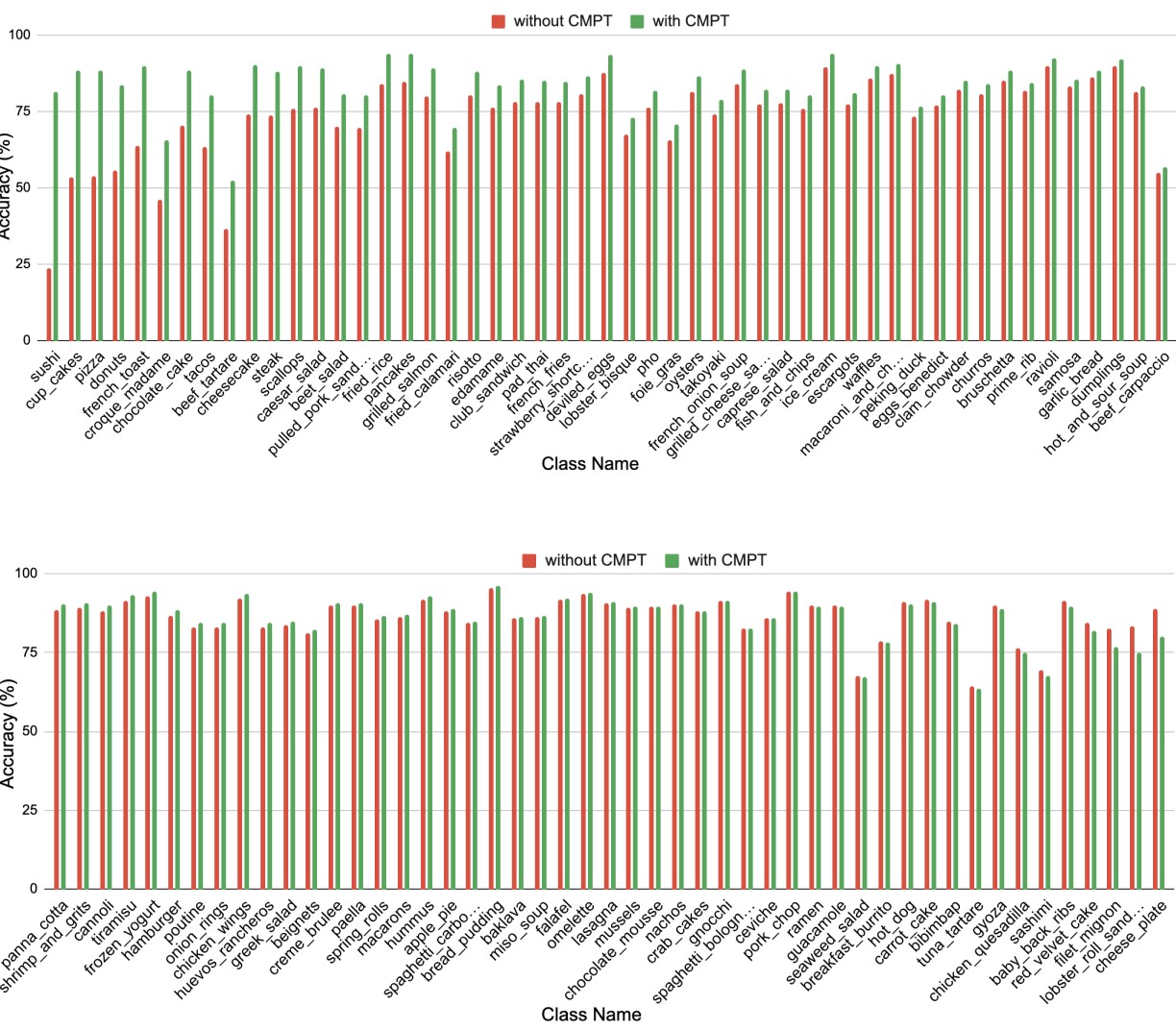

**Figure 9:** CMPTs improve performance across most of the classes when *image* modality is missing on UPMC Food-101 dataset. However, it fails to accurately predict certain classes (e.g., lobster_roll_sandwich/cheese_plate), likely due to the text modality's insufficient information for those classes. Classes are sorted based on the amount of performance improvement.

## A7   Do CMPTs Benefit All Classes?

To further evaluate CMPT's effectiveness, we conduct a per-class performance analysis on the UPMC Food-101 dataset, following the approach from Section 4.6.2.

As shown in Figure 8 and 9, CMPTs improve or maintain performance across most of the classes in both text-missing and image-missing scenarios. Notably, it mitigates performance degradation in modality-sensitive classes. For instance, when text is missing, classes such as *guacamole*, *pancakes*, and *foie_gras* experience severe drops in performance, but CMPTs effectively recovers much of the loss as shown in Figure 7b. Similarly, in image-missing scenarios, CMPTs provide substantial gains for *sushi*, *cup_cakes*, and *pizza* as shown in Figure 3a. However, a slight performance drop is observed in certain classes when either text or image is absent. For instance, CMPT fails to recover performance for classes such as *bibimbap* and *sashimi* in the absence of text, and for classes like *lobster_roll_sandwich* and *cheese_plate* when the image modality is missing. This is likely due to insufficient information in the available modality alone for those classes.

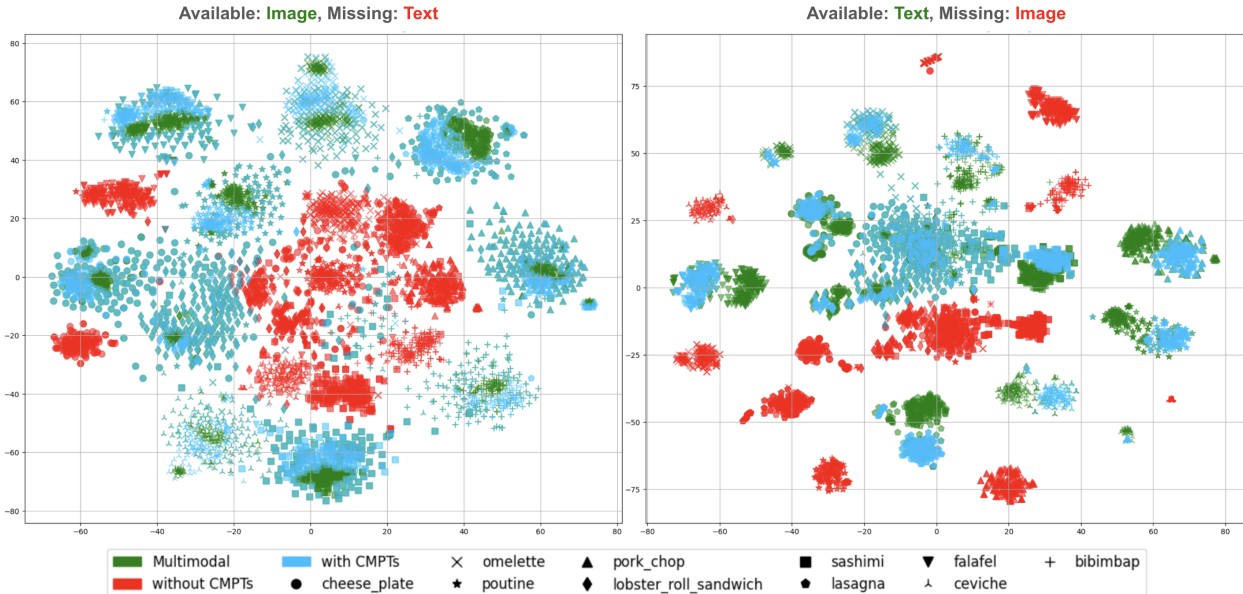

**Figure 10:** t-SNE plots of the fused feature tokens $\mathcal{T}$ for ten classes where the performance gains from CMPTs are limited. In these cases, the fused feature embeddings with CMPTs (blue) remain diffuse and overlapping, and exhibit weak alignment with the complete multimodal features (green). Although CMPTs show improvement over the baseline without CMPTs (red), the alignment remains insufficient to form consistently discriminative clusters, indicating limited or no improvement in missing modality robustness in these classes.

Overall, these results reinforce our previous findings, demonstrating that CMPTs enhance performance across most classes, leading to a strong overall performance improvement.

## A8 Limitations and Failure Case Analysis

While CMPTs demonstrate notable improvements in robustness, it is essential to understand their limitations. Figure 10 presents t-SNE plots of the fused feature tokens $\mathcal{T}$ for ten classes on the UPMC Food-101 dataset where the performance gains from CMPTs are limited. In these cases, the fused feature embeddings with CMPTs (blue) remain diffuse and overlapping, and exhibit weak alignment with the complete multimodal features (green). This is particularly evident for visually ambiguous classes or classes with high intra-class variance in the feature embedding space. Although CMPTs show improvement over the baseline without CMPTs (red), the alignment remains insufficient to form consistently discriminative clusters, indicating limited or no improvement in missing modality robustness.

We also analyze the attention maps from the last layer of the vision encoder for the CLS token and CMPTs in the text-missing scenario, as shown in Figure 11. The left half of the figure includes examples when the fused features with and without CMPTs provide correct predictions (approximately 60% samples show this pattern). For these cases, the CLS token already attends to semantically-relevant regions and make the correct prediction. CMPTs further refine the attention but do not change the model's prediction, as it is already correct. Thus, the inclusion of CMPTs does not necessarily improve performance. The right half of the figure shows examples when the fused features with CMPTs lead to incorrect predictions (approximately 3% samples show this pattern). We observe that CMPTs attend to semantically-relevant regions of the image but still lead to incorrect predictions. For example, in the *French Onion Soup* sample, the CMPT focuses heavily on the melted cheese surface—a visually salient but ambiguous feature—leading to a misclassification as *Shrimp and Grits*. Similarly, for the *Lobster Roll Sandwich*, the CMPT fails to recover from the original misclassification and predicts *Foie Gras* based on misleading visual cues. These examples highlight scenarios where either the available modality is insufficient or the learned cross-modal associations are misleading, resulting in CMPTs contributing to incorrect predictions rather than correcting them.

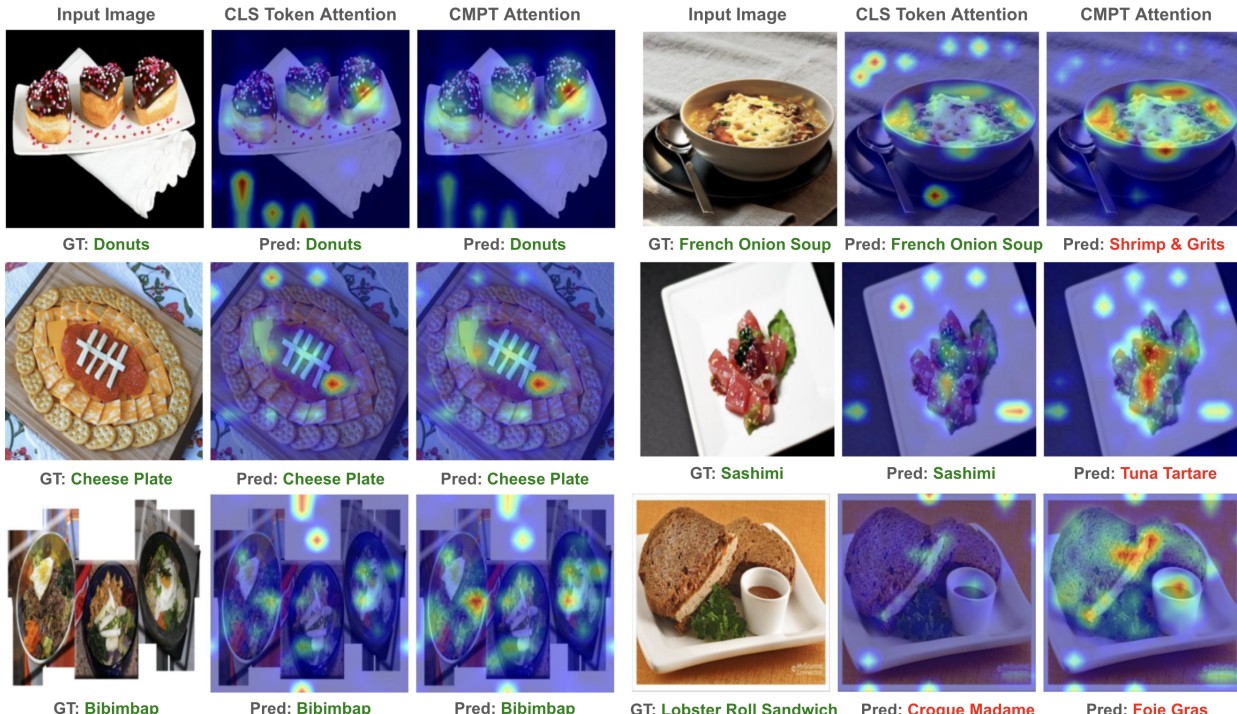

**Figure 11:** Attention map visualizations from the last layer of the vision encoder for the CLS token and CMPTs in the text-missing scenario on the UPMC Food-101 dataset. **Left:** Cases where the fused features with and without CMPTs provide correct predictions. Here, CLS tokens already attend to semantically-relevant regions and CMPTs provide further refinement, but do not change the prediction. **Right:** Cases where CMPTs attend to semantically-relevant regions but lead to incorrect predictions. The last example illustrates a scenario where both the fused features with and without CMPTs result in incorrect predictions. These examples highlight limitations of CMPTs when the available modality lacks sufficient discriminative information or when learned cross-modal associations are imprecise.

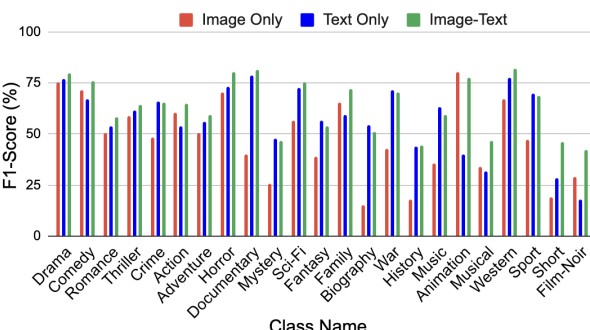

**Figure 12:** Per class performance analysis on MM-IMDb dataset. Certain classes show strong dependence on a specific modality.

## A9 Analysis of Modality Dependence

We observe that the impact of missing modalities varies significantly across classes. Certain classes show strong dependence on a specific modality. In this experiment, we train our model on modality-complete data and evaluate it on both modality-complete and modality-incomplete test samples.

Figure 12 presents per-class performance on the MM-IMDb dataset, revealing heterogeneous effects of missing modalities. For instance, the *Drama* and *Thriller* classes maintain consistent performance regardless of missing modalities. In contrast, the *Biography* and *Documentary* classes experience significant decline in

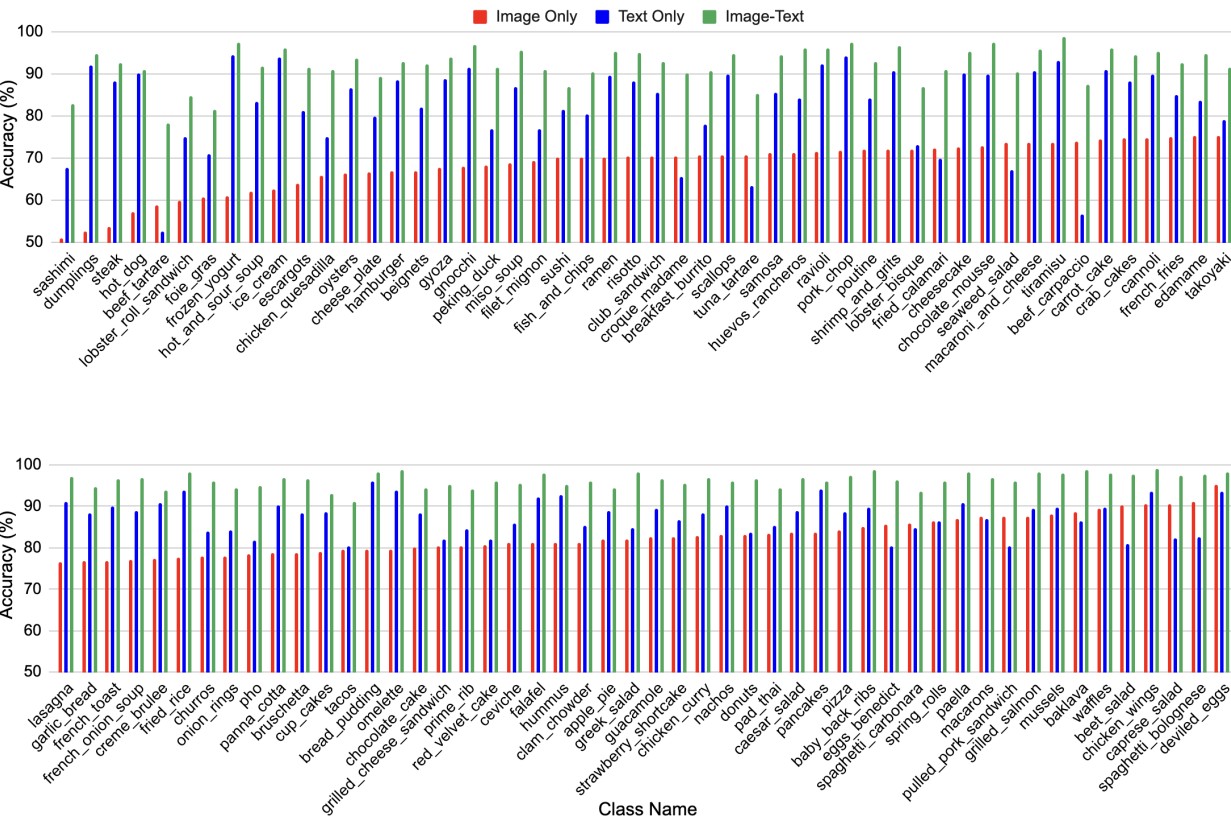

**Figure 13:** Per class performance analysis on UPMC Food-101 dataset. Certain classes show strong dependence on a specific modality.

performance when text is missing, even if images are available. Interestingly, certain classes like *Animation* show slightly higher performance with single-modality inputs, particularly in image-only scenario. Similar patterns emerge in *Sport* and *Music* categories, where text-only samples performs slightly better than complete modality. This discrepancy may stem from the simple sum-fusion strategy we use, which might not effectively model complex modality interactions. These findings indicate that the effect of missing modalities is not uniform across classes, with some being more modality-dependent than others.

Extending this analysis to UPMC Food-101, we observe that multimodal (image-text) data consistently outperforms unimodal inputs across all classes as shown in Figure 13. However, the reliance on specific modalities varies widely among food categories.

Classes such as *dumplings*, *Frozen_yogurt*, and *Ice cream* depend heavily on text, showing strong performance with text-only inputs but experiencing significant drops when text is missing. Conversely, categories like *tuna_tartare*, *beef_carpaccio*, and *beet_salad* rely more on image information, with performance deteriorating notably when images are absent. This observation suggests that some classes rely more on texts, while others depend heavily on images for accurate classification.

Interestingly, some classes such as *beef_tartare*, *chicken_quesadilla*, and *croque_madame* exhibit poor performance with unimodal inputs but improve significantly with both image and text, reinforcing the importance of multimodal data. Performance on these classes improve significantly when we utilize both image and text. Overall, these findings confirm that while multimodal information universally enhances performance, the degree of modality dependence remains heterogeneous across classes.

**Table 9:** Performance comparison with three different parameter efficient adaptation methods. We used rank=1 for all the methods. The best Accuracy (Acc.) for each scenario is highlighted in **bold**. Params (M) indicates the number of learnable parameters in millions.

| Datasets | Available Modality | Missing Modality | LoRA | | DoRA | | VeRA | |
|---|---|---|---|---|---|---|---|---|
| | | | Params (M) | Acc. (%) | Params (M) | Acc. (%) | Params (M) | Acc. (%) |
| UPMC Food-101 | Image | Text | | **75.66** | | 69.03 | | 70.82 |
| | Text | Image | 0.271 | **85.31** | 0.302 | 83.52 | 0.155 | 82.85 |
| | Image - Text | - | | **94.47** | | 92.59 | | 92.34 |
| AVE | Audio | Video | | 85.21 | | 86.57 | | **89.05** |
| | Video | Audio | 0.173 | 84.58 | 0.246 | **91.79** | 0.098 | 90.05 |
| | Audio - Video | - | | **96.77** | | 96.27 | | 96.27 |
| CREMA-D | Audio | Video | | **67.20** | | 67.07 | | 66.13 |
| | Video | Audio | 0.157 | **76.21** | 0.229 | 73.12 | 0.082 | 71.64 |
| | Audio - Video | - | | **88.84** | | 88.04 | | 86.02 |
| KS | Audio | Video | | 68.27 | | 68.77 | | **69.93** |
| | Video | Audio | 0.176 | 85.77 | 0.248 | **85.97** | 0.101 | 83.69 |
| | Audio - Video | - | | 91.21 | | **91.32** | | 90.71 |

## A10 Generalizability to Other Parameter-Efficient Adaptation Methods

To demonstrate that CMPTs are not restricted to a particular parameter-efficient adaptation method, we extend our analysis beyond LoRA (Hu et al., 2022), which is used in our primary experiments. Specifically, we evaluate CMPTs with two recent alternatives: DoRA (Liu et al., 2024) and VeRA (Kopiczko et al., 2024). For a fair comparison, we fix the rank to 1 across all methods. We train the models on modality-complete data and evaluate them on both modality-complete and modality-incomplete scenarios. The results are summarized in Table 9.

We find that the number of learnable parameters differs across methods, with VeRA being the most parameter-efficient and DoRA the least. In terms of performance, no single method consistently outperforms the others across all datasets. LoRA achieves the best overall results on the UPMC Food-101 and CREMA-D datasets, while DoRA shows superior performance on the Kinetics-Sound dataset and remains highly competitive on AVE. VeRA, while using substantially fewer parameters, delivers strong performance but is generally slightly below LoRA and DoRA, suggesting a trade-off between parameter efficiency and maximal performance.

Most importantly, our framework maintains robust performance in missing-modality scenarios regardless of the underlying adaptation method. The relative stability observed between complete and missing-modality settings is consistent across all three adaptation techniques. These findings confirm that CMPTs function as a modular approach whose effectiveness is not tied to any specific adaptation strategy, underscoring their scalability and versatility across different parameter-efficient methods.

## A11 Applicability to SigLIP-Based Models

To assess the architectural flexibility of our method, we replaced the ViT encoder (Dosovitskiy et al., 2021) with a pretrained SigLIP model (Zhai et al., 2023), which employs Global Attention Pooling (GAP) instead of a CLS token. For audio–video datasets, we retained the AST model as the audio encoder, and for image–text datasets, we used BERT as the text encoder. In this setup, CMPT tokens must perform a cross-architectural reconstruction: inferring missing audio or text features from SigLIP's vision embedding, and vice versa.

We train the models on modality-complete data and evaluate them on both modality-complete and modality-incomplete scenarios. The results in Table 10 show that our method is not restricted to a single encoder type. On UPMC Food-101 dataset, the SigLIP-based model consistently outperforms the ViT-based model, demonstrating that CMPTs can be effectively applied to GAP-based vision backbones for vision–language tasks. In contrast, for audio–video datasets, SigLIP exhibits sharp performance degradation when audio is missing. Here, CMPTs struggle to recover audio features from global video embeddings, likely because

**Table 10:** Performance comparison of our method with a ViT versus a SigLIP model as the vision backbone. The models are trained on complete multimodal data and tested on both complete and missing modality scenarios. We report % accuracy as the performance metric, and the highest score for each scenario is highlighted in **bold**.

| Datasets | Available Modality | Missing Modality | ViT (Dosovitskiy et al. (2021)) | SigLIP (Zhai et al. (2023)) |
|---|---|---|---|---|
| UPMC Food-101 | Image | Text | 75.66 | **77.50** |
| | Text | Image | 85.31 | **85.58** |
| | Image - Text | - | 94.47 | **94.58** |
| AVE | Audio | Video | 85.21 | **86.32** |
| | Video | Audio | **84.58** | 30.60 |
| | Audio - Video | - | **96.77** | 94.78 |
| CREMA-D | Audio | Video | **67.20** | 65.73 |
| | Video | Audio | **76.21** | 35.35 |
| | Audio - Video | - | **88.84** | 84.68 |
| KS | Audio | Video | 68.27 | **69.66** |
| | Video | Audio | **85.77** | 30.76 |
| | Audio - Video | - | **91.21** | 89.44 |

**Table 11:** Performance comparison of our method using *Base* versus *Large* ViT and BERT models. The models are trained on complete multimodal data and tested on both complete and missing modality scenarios. We report % accuracy for UPMC Food-101 and F1-Macro score for MM-IMDb dataset. Params (M) indicates the number of learnable parameters in millions. The best performance for each scenario is highlighted in **bold**.

| Datasets | Available Modality | Missing Modality | ViT-Base and BERT-Base | | ViT-Large and BERT-Large | |
|---|---|---|---|---|---|---|
| | | | Params (M) | Performance | Params (M) | Performance |
| MM-IMDb (F1-Macro) | Image | Text | | 46.97 | | **53.42** |
| | Text | Image | 0.211 | 56.32 | 0.540 | **59.92** |
| | Image - Text | - | | 63.58 | | **64.00** |
| UPMC Food-101 (Accuracy) | Image | Text | | 75.66 | | **77.17** |
| | Text | Image | 0.271 | **85.31** | 0.501 | 83.91 |
| | Image - Text | - | | **94.47** | | 93.74 |

GAP discards the fine-grained temporal information that is critical for inferring a corresponding audio representation. The reverse process, however, is more successful. When the video modality is missing, AST's audio features provide a strong basis for reconstructing SigLIP's global video representation. This suggests that AST's feature space is more compatible with SigLIP than vice versa.

Overall, these findings indicate that CMPTs remain applicable to GAP-based models like SigLIP, but their effectiveness depends on the modality. While GAP facilitates efficient vision embeddings, it limits the representational richness required for audio reconstruction, constraining CMPTs in certain cases. Nonetheless, the results confirm that CMPTs are not inherently incompatible with SigLIP, though their utility is shaped by architectural and modality-specific factors.

## A12 Scalability with Larger Vision and Text Encoders

To assess the scalability of our framework with larger encoders, we replaced the ViT-Base and BERT-Base encoders used in our primary experiments with ViT-Large and BERT-Large. We train the models on modality-complete data and evaluate them on both modality-complete and modality-incomplete scenarios on the UPMC Food-101 and MM-IMDb datasets.

The results, summarized in Table 11, show that our framework can leverage the increased capacity of larger backbones. On MM-IMDb, scaling yields consistent improvements in F1-Macro across all scenarios.

The gains are particularly notable in missing-modality cases, with a +6.45% increase when text modality is missing and +3.60% when image modality is missing. These results suggest that stronger backbone representations improve CMPT's ability to reconstruct missing features and maintain robust predictions.

On UPMC Food-101, the larger backbone improves image-only accuracy by +1.51% but shows a decline of -1.41% for text-only performance. With both modalities available, performance remains comparable (94.47 vs. 93.74). This indicates a potential trade-off, where the added capacity may overfit to the text modality, reducing generalization when the image is absent.

In summary, CMPT scales effectively with larger backbones, particularly on complex datasets such as MM-IMDb. At the same time, the results highlight that the benefits of scaling depend on dataset and modality characteristics, underscoring the need for careful backbone selection to balance capacity and generalization.

