# OpenReview forum: "Robust Multimodal Learning via Cross-Modal Proxy Tokens"
_TMLR — Accepted by TMLR_

### Review · Reviewer_SQkA · 2025-06-29

**Summary Of Contributions:**

This paper introduces Cross-Modal Proxy Tokens (CMPTs) as a parameter-efficient mechanism to enhance the robustness of multimodal models in scenarios involving missing modalities. CMPTs serve as surrogates for missing modality class tokens, learned via attention over available modality tokens and supervised through an alignment loss. The framework builds on pretrained unimodal transformers, adapted using lightweight low-rank adapters (LoRA), and integrates a gating mechanism for selective fusion. The authors evaluate the method across five datasets spanning image-text and audio-video tasks and benchmark against a comprehensive set of state-of-the-art methods. The results demonstrate consistent gains under both complete and partial modality settings.

The key contributions include:
- A simple yet effective method for missing modality compensation via proxy token learning.
- Robust performance across datasets and tasks, with empirical superiority in both full and partial input scenarios.
- Extensive experiments including ablation, class-wise analysis, and generalization under varying missing ratios.

**Audience:**

Yes

**Broader Impact Concerns:**

If training data embeds biased correlations (e.g., gendered visual-textual links such as associating certain professions with specific genders or linking emotional expression to particular demographics), CMPTs may propagate or amplify these, especially in missing modality cases where the full context is unavailable to correct such biases.

**Claims And Evidence:**

Yes

**Requested Changes:**

1. Add qualitative analyses such as attention visualizations or embedding projections to show what CMPTs “attend to” and how closely they resemble the original CLS tokens of the missing modality, and show token-level dynamics in varying input conditions (e.g., which modality dominates proxy behavior).
2. Evaluate in multi-modal (>2) settings. Add at least one tri-modal experiment (e.g., audio-video-text), even if synthetic or partial, to test scalability and fusion behavior.
3. Include a comparison against partially fine-tuned models (e.g., unfrozen last layers of the encoder) to clarify how much of CMPT’s gain comes from structure vs. just adaptation strategy.
4. Discuss failure modes more explicitly where performance drops on certain classes or modality losses are not recoverable, including a dedicated discussion or error analysis table to help practitioners make informed deployment decisions.

**Strengths And Weaknesses:**

[S1] The idea of a proxy token as a stand-in for missing modality representations is easy to implement, avoiding the need for complex imputation.

[S2] By employing frozen encoders and rank-1 LoRA adapters, the method keeps the learnable parameter overhead very low.

[S3] The experimental section is extensive, featuring comparisons across different missing configurations (training/inference), strong baselines, generalization tests, and per-class impact analysis.

[S4] The method maintains strong predictive performance even as modality availability degrades.

---

[W1] While the empirical results are well-demonstrated, the paper lacks a deeper theoretical or conceptual analysis of why the CMPTs effectively capture the semantics of missing modalities. Are they learning structured embeddings, redundancy, or co-occurrence priors?

[W2] Scalability beyond two modalities is unclear. All experiments are limited to bimodal settings. Extending the method to three or more modalities (e.g., image, audio, text) introduces ambiguity in token alignment, gating, and fusion, yet this is not addressed.

[W3] While dependence on pretrained encoders is efficient, this dependence also limits flexibility. The frozen encoder assumption may not hold in niche or low-resource modalities/domains where pretrained models are unavailable or poorly aligned.

[W4] Despite overall improvements, certain classes (e.g., documentary, short) degrade significantly in the absence of text. This suggests that CMPTs struggle to extrapolate for semantically sparse or modality-specific categories.

---

> ### Author Response · Authors · 2025-07-28
> **Response to the Comments from Reviewer SQkA (1/3)**
>
> We sincerely thank Reviewer SQkA for their thoughtful, detailed, and positive feedback. We are encouraged by the reviewer’s recognition of our method as a **simple yet effective approach** with **robust performance** and **extensive experiments**. Below, we address the weaknesses (W) and requested changes (RC) in detail:
>
> **1. W1 and RC1: Deeper and Qualitative Analysis:**
>
> We appreciate the reviewer’s insightful suggestion. In the revised manuscript, we have added detailed qualitative analyses, including both fused feature projections and attention visualizations (in **Section 4.6.4 and A8**). **Figure 5** presents t-SNE plots of the fused feature tokens $\mathcal{T}$ for ten classes (marker shape) under different modality settings (marker color). The t-SNE plots show that fused features without CMPTs (red) deviate significantly from the complete multimodal features (green). Incorporating CMPTs (blue) provides embeddings that closely align with the complete multimodal features, indicating improved semantic alignment and robustness.
>
> We also include attention map visualizations for the CLS token and CMPTs in the missing-text scenario in **Figure 6**. We selected random examples from six classes where fused features without CMPTs provide incorrect predictions, but including the CMPTs leads to correct predictions (approximately 15% samples in the dataset show this pattern). The attention maps show that CLS tokens largely attend to background or semantically-irrelevant regions, which likely leads to incorrect predictions. In contrast, CMPTs attend to semantically-relevant regions for the respective classes and lead to accurate predictions. These visualizations demonstrate that CMPTs not only act as proxies for the missing modality but also reshape the model’s attention to reflect modality-relevant cues. For the sake of completeness, we have also included attention maps for some cases when the fused features with and without CMPTs provide correct predictions (approximately 60% samples show this pattern) and when the fused features with CMPTs seemingly attend to relevant regions but lead to incorrect predictions (approximately 3% samples show this pattern) in **Figure 11**. We note that the selected images seem challenging for human observers as well.
>
> We hope these results provide the additional insight requested by the reviewer.
>
> **2. W2 and RC2: Scalability Beyond Two Modalities:**
>
> As discussed in **Section 5 (Limitations and Future Directions)**, this paper focuses on two-modality settings, primarily due to the availability of standardized benchmarks and established baselines that allow for fair and direct comparisons. We agree that evaluating CMPT in higher-order settings (with >2 modalities) is a valuable direction to assess its scalability and broader applicability. We are currently working on a general framework, but believe such an extension and comprehensive evaluations are beyond the scope of this paper.

---

> ### Author Response · Authors · 2025-07-28
> **Response to the Comments from Reviewer SQkA (2/3)**
>
> **3. W3 and RC3: Dependence on Pretrained Encoders and Comparison to Partial Finetuning:**
>
> We thank the reviewer for raising this important point regarding encoder flexibility and the source of CMPT’s performance improvements. We address the three key aspects of the comment below.
>
> * **Frozen Encoder Assumption and Model Flexibility:** While we start with pretrained unimodal encoders (e.g., ViT, BERT, AST), our method adds Low-Rank Adaptation (LoRA) modules within each encoder layer, as described in **Section 3.3**. In our setup, *although the base encoder weights remain fixed, the LoRA modules offer flexibility comparable to partial fine-tuning, allowing the model to update its internal representations for robust multimodal learning.* We have also added an ablation study in **Section A4** showing that LoRA with r = 1 is sufficient to capture essential task-specific information and provides an optimal balance, delivering robust performance with the greatest parameter efficiency.
> * **Niche/Low-Resource Modalities:** Our framework is designed for and evaluated on commonly used modalities—vision (image and video), text, and audio—for which high-quality pretrained encoders (such as ViT, BERT, and AST) are widely available. We agree that in niche or low-resource domains, pretrained encoders may be unavailable or suboptimal. In such cases, a pretraining step would be necessary to initialize the modality-specific encoder. This is a standard practice in many real-world machine learning pipelines. Once a modality-specific encoder is available, our framework can be seamlessly applied.
> * **Comparison Against Partially Fine-tuned Models:** We appreciate the reviewer’s request to isolate CMPT’s structural benefits from its adaptation strategy. In **Section 4.6.1**, the model referred to as *baseline* employs pretrained unimodal encoders with LoRA (rank = 1), enabling parameter-efficient adaptation while keeping the encoder weights frozen. *This can be considered as a partially fine-tuned model which provides a fair and flexible comparison point.* We further evaluate:
>     * A modality dropout model, which uses the same LoRA fine-tuning strategy but introduces random modality dropout during training for robustness.
>     * Our proposed CMPT, which builds upon this fine-tuning setup and adds cross-modal proxy tokens to approximate missing modality representations.
>
> As shown in **Figure 3**, the baseline model suffers from significant performance drops when modalities are missing. Modality dropout helps to a degree, but *CMPT consistently outperforms both alternatives achieving a +32.06% gain over the baseline and a +11.55% gain over modality dropout in the missing-image scenario on MM-IMDb dataset (**Figure 3a**)*. Similar trends are observed for missing-text scenarios (**Figure 3b**) and across datasets, as discussed in **Section A6**.
>
> These results demonstrate that CMPT’s performance improvements are not solely due to increased model flexibility or fine-tuning strategy. Rather, they arise from its explicit structural design for cross-modal approximation of missing information.
> We hope this clarifies the design choices and empirical contributions of CMPT. We are happy to further elaborate or include additional details if the reviewer believes it would strengthen the manuscript.

---

> ### Author Response · Authors · 2025-07-28
> **Response to the Comments from Reviewer SQkA (3/3)**
>
> **4. W4 and RC4: Failure Modes:**
>
> We thank the reviewer for highlighting the importance of discussing failure modes more explicitly. We provided a detailed per-class performance analysis in **Section 4.6.2** and **Appendix A7**. We have revised these sections to improve the clarity and emphasis of the failure case discussion. *We have also added fused feature projections and attention map visualizations to further analyze failure cases in detail in Section A8*.
>
> * Specifically, **Figure 4** presents per-class performance with and without CMPTs on the MM-IMDB dataset. As discussed in **Section 4.6.2**, when text is missing, CMPTs are able to recover performance across nearly all classes except for documentary and short, where degradation remains. *The figure caption explicitly notes that the model "fails to accurately predict certain classes (e.g., short/documentary) in the absence of text."* In the image-missing scenario, CMPTs provide improvements across all classes.
> * Similarly, **Figures 8 and 9** show per-class performance with and without CMPTs on the UPMC Food-101 dataset. These results indicate that CMPTs improve or maintain performance in most classes for both text- and image-missing settings. However, for a few classes, performance degradation remains unrecoverable. For example, *CMPTs fail to fully recover performance for classes such as bibimbap and sashimi when text is missing, and lobster_roll_sandwich and cheese_plate when image is missing.* These observations are discussed in *Section A7*. Given the large number of classes (101) in this dataset, we highlight representative examples in the text while referring readers to the full visual breakdown in the figures.
> * Additionally, we have included qualitative analysis in **Section A8**, where we discuss these cases more thoroughly. In **Figure 10**, we show that in certain classes, the fused feature embeddings with CMPTs (blue) remain diffuse and overlapping, exhibiting weak alignment with the complete multimodal features (green). Although CMPTs show some improvement over the baseline without CMPTs (red), the alignment remains insufficient to form consistently discriminative clusters. Similarly, in **Figure 11**, we show that while CMPTs refine attention to focus on semantically-relevant regions, they can still lead to incorrect predictions in cases where the available modality is insufficient or the learned cross-modal associations are misleading.
>
> We hope these clarifications and revisions sufficiently address the reviewer’s concern. However, we would be happy to further expand the discussion or include a dedicated summary table if the reviewer feels it would strengthen the manuscript.
>
> We have updated the manuscript incorporating the changes. **We have included an annotated manuscript with all changes highlighted in the supplementary materials.** We sincerely hope that these revisions address the reviewer’s constructive suggestions and contribute to improving the quality of the paper. We are grateful for the reviewer’s time, thoughtful feedback, and valuable insights.

---

> ### Comment · Action_Editor_rzFx · 2025-09-04
>
> Dear reviewer
>
> Please submit your final recommendation in light of the rebuttal and other reviews

---

> > ### Comment · Reviewer_SQkA · 2025-09-08
> >
> > Thank you to the authors for the detailed rebuttal and revisions. I appreciate the substantial effort to address my earlier concerns and to strengthen the paper. That said, I still see some limitations that prevent me from being fully convinced about the paper’s readiness for TMLR in its current form:
> > 1. The dependence on strong pretrained models is both a strength and a weakness. The rebuttal clarified how LoRA provides flexibility, but the broader applicability in low-resource or niche modalities remains somewhat underexplored.
> > 2. I appreciate the inclusion of per-class breakdowns and qualitative visualizations, but the failure cases (e.g., documentary/short genres, bibimbap/sashimi in Food-101) highlight important limitations in CMPT’s ability to recover semantically sparse categories. I would encourage the authors to bring these insights even more to the forefront of the discussion, as they are valuable for practitioners.
> >
> > On the positive side, the paper is well written, easy to follow, and proposes a simple yet effective mechanism that consistently improves robustness under missing modality scenarios. The experimental evaluation is solid across several benchmarks, and the qualitative analyses now provide more intuition about what CMPTs are learning.
> >
> > I lean toward weak accept. While the novelty is incremental and the scope somewhat narrow (two-modality setting, one adaptation method), I believe the work is technically sound and well-motivated.
> >
> > For future iterations, I strongly encourage the authors to: 1) extend the framework to tri-modal or higher-order settings, and 2) experiment with other adaptation strategies beyond LoRA to better establish the generality of the method.

---

> ### Author Response · Authors · 2025-09-15
> **Experiments with other adaptation strategies beyond LoRA**
>
> We sincerely thank the reviewer for the thoughtful follow-up and for recognizing the strengths of our work, including the clarity of writing, robustness improvements under missing modalities, and the value of our qualitative analyses. We also appreciate the constructive feedback on the limitations and future directions.
>
> * **On broader generality beyond LoRA:** To demonstrate that CMPTs are not restricted to a particular parameter-efficient adaptation method, we extended our analysis beyond LoRA, which was used in our primary experiments. **Specifically, we evaluated CMPTs with two recent alternatives: DoRA [1] and VeRA [2].** For a fair comparison, we fixed the rank to 1 across all methods. Models were trained on modality-complete data and evaluated on both modality-complete and modality-incomplete scenarios. The results are summarized in Table R3T1.
>
> **R3T1: Experiments with different parameter-efficient adaptation methods.**
>
> | Datasets | Available Modality | Missing Modality | LoRA Params (M) | LoRA Accuracy (%) | DoRA Params (M) | DoRA Accuracy (%) | VeRA Params (M) | VeRA Accuracy (%) |
> |:---:|:---:|:---:|:---:|:---:|:---:|:---:|:---:|:---:|
> | UPMC Food-101 | Image | Text | 0.271 | **75.66** | 0.302 | 69.03 | 0.155 | 70.82 |
> | | Text | Image | | **85.31** | | 83.52 | | 82.85 |
> | | Image - Text | - | | **94.47** | | 92.59 | | 92.34 |
> | AVE | Audio | Video | 0.173 | 85.21 | 0.246 | 86.57 | 0.098 | **89.05** |
> | | Video | Audio | | 84.58 | | **91.79** | | 90.05 |
> | | Audio - Video | - | | **96.77** | | 96.27 | | 96.27 |
> | CREMA-D | Audio | Video | 0.157 | **67.20** | 0.229 | 67.07 | 0.082 | 66.13 |
> | | Video | Audio | | **76.21** | | 73.12 | | 71.64 |
> | | Audio - Video | - | | **88.84** | | 88.04 | | 86.02 |
> | KS | Audio | Video | 0.176 | 68.27 | 0.248 | 68.77 | 0.101 | **69.93** |
> | | Video | Audio | | 85.77 | | **85.97** | | 83.69 |
> | | Audio - Video | - | | 91.21 | | **91.32** | | 90.71 |
>
> We find that the number of learnable parameters differs across methods, with VeRA being the most parameter-efficient and DoRA the least. In terms of performance, no single method consistently outperforms the others across all datasets. LoRA achieves the best overall results on the UPMC Food-101 and CREMA-D datasets, while DoRA shows superior performance on the Kinetics-Sound dataset and remains highly competitive on AVE. VeRA, while using substantially fewer parameters, delivers strong performance but is generally slightly below LoRA and DoRA, suggesting a trade-off between parameter efficiency and maximal performance.
>
> **Most importantly, our framework maintains robust performance in missing-modality scenarios regardless of the underlying adaptation method. The relative stability observed between complete and missing-modality settings is consistent across all three adaptation techniques.** These findings confirm that CMPTs function as a modular approach whose effectiveness is not tied to any specific adaptation strategy, underscoring their scalability and versatility across different parameter-efficient methods. We discuss these findings in **Section A10** of the updated manuscript.
>
> * **On tri-modal and higher-order extensions:** We agree that evaluating CMPT in higher-order settings (with more than two modalities) is an important direction to further assess its scalability and broader applicability. As discussed in Section 5 (Limitations and Future Directions), we are actively investigating this as part of our ongoing work and plan to report detailed findings in future publications.
>
> **In addition, we have added experiments with SigLIP-based models [3] as the vision encoder (Section A11) and with larger vision and text encoders (Section A12). An annotated manuscript with all changes highlighted is included in the supplementary materials.** We sincerely hope the current revisions demonstrate the scalability and generality of our approach across encoder sizes and adaptation techniques. We thank the reviewer once again for the constructive feedback and for leaning toward acceptance.
>
> **References**
>
> [1] Liu, Shih-Yang, et al. "Dora: Weight-decomposed low-rank adaptation." Forty-first International Conference on Machine Learning. 2024.
>
> [2] Kopiczko, Dawid J., Tijmen Blankevoort, and Yuki M. Asano. "Vera: Vector-based random matrix adaptation." arXiv preprint arXiv:2310.11454 (2023).
>
> [3] Zhai, Xiaohua, et al. "Sigmoid loss for language image pre-training." Proceedings of the IEEE/CVF international conference on computer vision. 2023.

---

> > ### Comment · Reviewer_SQkA · 2025-09-15
> >
> > I appreciate the authors for their substantial effort in conducting additional experiments. I will maintain my position of accepting this paper.

---

### Review · Reviewer_Th7k · 2025-07-14

**Summary Of Contributions:**

The paper focuses on multimodal learning in scenarios of missing modalities. It proposes cross-modal proxy tokens (CMPTs) to learn an approximation of the missing modality class tokens from the available modality using an alignment MSE loss. The paper applies LoRA to frozen unimodal encoders to learn the CMPTs.

**Audience:**

Yes

**Broader Impact Concerns:**

No concerns.

**Claims And Evidence:**

Yes

**Requested Changes:**

Critical for acceptance:
0- please justify the usage of  lambda = 0.2 in Sec 3.4. Ideally, authors would include a Table sweeping lambda in [0, some value) for one of the basic comparisons. Ideally, this would be carried out for 2 datasets, to show if the choice of lambda is critical and/or very dataset-dependent.


Other optional comments to strengthen the work:
1- As mentioned in Weaknesses, it would be a service to the community if authors paved the way for more comprehensive evaluations. Specifically, a) reporting results for different missing rates of images in Sec 4.5.; b) IT'd be great if authors included an additional dataset in at least one of their studies, eg the larger and more recent VGGSound. For these two cases, authors cannot compare to previous work, but authors can set a baseline for future works.
2- it would be greatly appreciated to report results using mean & stdev (as in Table 1) in at least some of the other Tables too.
3- in Fig 1, consider inverting the arrows feeding the alignment loss such that they reflect CMPT_1 vs CLS_2 and vice versa
4- seeing as the considered tasks are eminently classification, perhaps this can be mentioned earlier in the text, eg Sec 3.4, or even in the Introduction.
5- authors state: "LoRA with rank 1 is sufficient to achieve better performance compared to existing state-of-the-art methods". Have authors considered other ranks? Are further boosts observable? Please provide more context.
6- in 4.3.1 it would be prudent to explicitly acknowledge the fact that the proposed system works comparable to that of Kim & Kim when reporting UPMC Food results (in two conditions).
7- in 4.3.2, maybe first two sentences are redundant?

**Strengths And Weaknesses:**

Strengths
Paper is well written and explanations are clear. While the proposed method does not seem extremely novel, its simplicity and elegance are appreciated (as well as its strong performance). The experimental validation seems solid: extensive experiments across multiple datasets and scenarios, as well as reporting mean & stdev (though it seems this is only reported in Table 1?).


Weaknesses
- The choice of lambda = 0.2 in Sec 3.4 is not justified at all. This warrants more context.
- While I understand the adoption of benchmarks used by previous work, the selection of AudioVisual datasets seems a bit weak, with AVE and CREMA-D being pretty small, and KS not much larger. It'd be great if authors included an additional dataset in their study, eg the larger and more recent VGGSound.

---

> ### Author Response · Authors · 2025-07-29
> **Response to the Comments from Reviewer Th7k (1/2)**
>
> We sincerely thank Reviewer Th7k for their thorough, insightful, and constructive review. We are encouraged that the reviewer found our method to be **simple and elegant**, and our experiments to be **extensive across multiple datasets and scenarios**. Below, we address the weaknesses and requested changes in detail:
>
> **0. The Choice of λ=0.20:**
>
> We thank the reviewer for raising this important point regarding the choice of the hyperparameter λ. We have conducted an ablation study on two datasets (UPMC Food-101 and AVE) by *varying λ in the range [0.0, 0.5]*.
>
>
> **Table R2T1: Ablation study on the alignment loss weight λ.**
> | Datasets | Available Modality | Missing Modality | λ=0.0 | λ=0.10 | λ=0.20 | λ=0.30 | λ=0.40 | λ=0.50 |
> | :--- | :--- | :--- | :---: | :---: | :---: | :---: | :---: | :---: |
> | UPMC Food-101 | Image | Text | 73.12 | 73.48 | **75.66** | 73.72 | 74.39 | 74.46 |
> | UPMC Food-101 | Text | Image | 84.31 | 84.79 | **85.31** | 84.51 | 84.81 | 84.88 |
> | UPMC Food-101 | Image-Text | - | 93.96 | 94.14 | **94.47** | 94.13 | 94.34 | 94.21 |
> | AVE | Audio | Video | 84.82 | 84.33 | **85.21** | 84.82 | 84.33 | 84.82 |
> | AVE | Video | Audio | 82.83 | 83.08 | 84.58 | **85.32** | 81.84 | 84.83 |
> | AVE | Audio-Video | - | 95.77 | 96.02 | **96.77** | 96.52 | 96.52 | 96.27 |
>
>
> The results, presented in **Table R2T1**, indicate that λ=0.20 demonstrates superior or near-optimal performance consistently across both datasets. For this reason, we set λ=0.20 for all our experiments, as it provides a balance in performance across different modality conditions. We added **Section 4.6.3** to the revised manuscript, providing a detailed analysis and corresponding results in **Table 5**. We hope this addition helps clarify the rationale for our choice and strengthens the paper.
>
> **1 (a). Results for Different Missing Rates of Images:**
>
> We agree that reporting results on different missing rates of images is a valuable contribution for establishing future benchmarks. We reported an analysis on this in **Section 4.6.1 (Figure 3a)** for the MM-IMDb dataset and in **Appendix A6 (Figure 7a)** for the UPMC Food-101 dataset.
>
> As the reviewer noted, standard benchmarks for this specific setup are not available. To address this and create a baseline for future works, we compared our method against two relevant baselines: (1) a standard “baseline” model trained without modality dropout and CMPT and (2) a model trained with modality dropout for missing modality robustness. *Our findings show that CMPTs significantly outperform both of these baselines.* Notably, CMPTs become increasingly effective as the amount of missing data increases, demonstrating its strong ability to substantially mitigate performance degradation in the presence of severe modality loss. We hope this correctly addresses the suggestion.
>
> **1 (b). Experiments on VGGSound Dataset:**
>
> We agree that evaluating our method on a large-scale dataset like VGGSound would be a valuable and interesting extension. However, given the substantial computational resources and engineering effort required for experiments on a dataset of this magnitude (over 200,000 videos), we believe that such an extensive evaluation falls beyond the scope of the current manuscript. We have noted this as a promising direction for future research. We hope that the evidence provided from the five diverse datasets already in the study is sufficient to demonstrate the core contributions and effectiveness of our method.
>
> **2. Reporting Mean and Standard Deviation:**
>
> We thank the reviewer for this suggestion and for the opportunity to clarify our reporting methodology.
>
> For each set of experiments, we train one model and then evaluate it on different available and missing modality configurations. Our reporting strategy for these evaluations is directly tied to whether the testing configuration is stochastic or deterministic.
> * **For Tables 1 and 8 (Stochastic Setups):** These experiments involve random sampling for partially missing modalities (e.g., 65% image / 65% text). Because the specific subset of data with missing modalities is chosen randomly for each evaluation, we conduct the tests over three independent runs with different seeds to account for this sampling variance. For these results, we report the mean and standard deviation.
> * **For Tables 2, 3, and 4 (Deterministic Setups):** In contrast, these scenarios involve either a complete modality being absent or all modalities being present. As there is no random sampling of data subsets in these cases, the evaluation is deterministic. The result for the trained model is therefore fixed and reproducible.
>
> We hope this explanation fully clarifies the distinction in our reporting method. However, we would be happy to run additional experiments and report these statistics if it is necessary to improve the paper.

---

> ### Author Response · Authors · 2025-07-29
> **Response to the Comments from Reviewer Th7k (2/2)**
>
> **3. Inverting the Arrows in Figure 1 (Model Diagram):**
>
> We have revised **Figure 1** by incorporating *color-coded arrows leading to the alignment loss block*. This change is intended to more clearly illustrate the alignment of $CMPT_1$ with $CLS_2$ and vice versa. We hope this change adequately addresses the concern. We are happy to make further adjustments if necessary.
>
> **4. Mentioning Classification Tasks in Section 3.4:**
>
> We have specified that our work focuses on classification tasks in **Section 3.4** before Equation 7 to provide better context for the reader.
>
> **5. Ablation Study Related to LoRA Rank:**
>
> To determine the optimal LoRA rank, we conducted an ablation study evaluating ranks $r \in$ {1, 2, 4}. For this analysis, we trained our models on modality-complete data and evaluated their performance in both complete and missing modality scenarios
>
>
> **Table R2T2: Ablation study on LoRA rank.**
>
> | Datasets | Available Modality | Missing Modality | r=1 Params (M) | r=1 Acc. (%) | r=2 Params (M) | r=2 Acc. (%) | r=4 Params (M) | r=4 Acc. (%) |
> | :--- | :--- | :--- | :---: | :---: | :---: | :---: | :---: | :---: |
> | UPMC Food-101 | Image | Text | 0.271 | **75.66** | 0.376 | 75.62 | 0.671 | 70.38 |
> | UPMC Food-101 | Text | Image | | 85.31 | | **85.61** | | 83.92 |
> | UPMC Food-101 | Image-Text | - | | 94.47 | | **94.64** | | 93.30 |
> | AVE | Audio | Video | 0.173 | **85.21** | 0.319 | 85.07 | 0.614 | 84.33 |
> | AVE | Video | Audio | | **84.58** | | 84.30 | | 80.84 |
> | AVE | Audio-Video | - | | **96.77** | | **96.77** | | 96.27 |
> | CREMA-D | Audio | Video | 0.157 | **67.20** | 0.303 | 65.32 | 0.598 | 65.46 |
> | CREMA-D | Video | Audio | | 76.21 | | 75.54 | | **76.61** |
> | CREMA-D | Audio-Video | - | | **88.84** | | 87.63 | | 87.77 |
> | KS | Audio | Video | 0.176 | 68.27 | 0.322 | **69.93** | 0.617 | 69.54 |
> | KS | Video | Audio | | 85.77 | | 85.20 | | **86.43** |
> | KS | Audio-Video | - | | **91.21** | | 91.06 | | 90.59 |
>
>
> Table **R2T2** shows performance under both complete and missing modality conditions for different LoRA ranks. *Our results show that a rank of r=1 provides robust performance across the majority of scenarios while being the most parameter-efficient*. We found that while higher ranks increase parameter count, they did not yield consistent performance improvements and, in some cases, led to a slight decrease in performance, suggesting that a larger number
> of learnable parameters may not be necessary for these tasks and could introduce a risk of overfitting.
>
> In the updated manuscript, we added **Section A4** with a detailed discussion, and **Table 7** presents the corresponding results for context. We believe this analysis strengthens the paper and justifies our choice of using rank 1.
>
> **6. Explicitly Acknowledging Kim & Kim in Section 4.3.1:**
>
> We have updated **Section 4.3.1** to acknowledge the comparable performance of Kim & Kim on the UPMC Food-101 dataset in two cases.
>
> **7. Removing Redundant Sentence from Section 4.3.2:**
>
> We have revised the first two sentences to remove the redundancy.
>
> **8. Additional Update:**
>
> We also wish to correct a typographical error in our original manuscript. In **Table 2**, for the UPMC Food-101 dataset in image-missing scenario, our method's performance was incorrectly listed as 80.66. The correct value is 85.31, as shown in **Figure 7(a)**. We apologize for this oversight and have corrected the table in the revision.
>
> We have revised the manuscript to incorporate these valuable feedbacks. **We have included an annotated manuscript with all changes highlighted in the supplementary materials.** We are grateful for the time and thoughtful feedback. We believe that the revisions have meaningfully improved the clarity and quality of our paper, and we hope they satisfactorily address all concerns. We thank the reviewer once again for the valuable insights and careful review.

---

> ### Comment · Action_Editor_rzFx · 2025-09-04
>
> Dear reviewer
>
> Please submit your final recommendation in light of the rebuttal and other reviews

---

### Review · Reviewer_ZCWM · 2025-08-02

**Summary Of Contributions:**

The paper presents a method for low-adaption multimodal classification under missing modalities. The method proposes to use a new token CMPT for each modality that can learn to predict CLS token of the missing modality. The encoder parameters are updated using LoRA. The paper presents SOTA results on different datasets including audiovisual and text-images.

**Audience:**

No

**Claims And Evidence:**

Yes

**Requested Changes:**

see Weaknesses above

**Strengths And Weaknesses:**

Strengths:

-	The method is simple and effective.
-	The paper is well written and easy to follow.

Weaknesses:

-	The method uses pretrained encoders, creating a new classifier based on the CLS tokens of the modalities. How would this method work if there is already a multimodal classifier trained?
-	The experiments are performed using ViT-B from CLIP. How does the method work on bigger architecture? How about architectures such as SigLIP that does not have a CLS token?
-	There is a missing related work that is very similar (applied on different domain) to this work, MoRA [A] that was not discussed in the related work.
-	Why was LoRA chosen when there are many other low-rank adaptation methods in the literature (eg. DoRA, VeRA etc)
-	How would this method work with more modalities such as 3?
-	Since the method is straightforward, more experiments are needed to improve the relevance of the paper.

[A] MoRA: LoRA Guided Multi-Modal Disease Diagnosis with Missing Modality, MICCAI 2024

---

> ### Author Response · Authors · 2025-08-06
> **Response to the Comments from Reviewer ZCWM (1/2)**
>
> We thank the reviewer for their time and for providing a thoughtful and constructive review of our manuscript. We are encouraged that the reviewer found our method to be **simple and effective** and the paper **well written and easy to follow**.
>
> We have carefully considered the weaknesses and requested changes. Below, we address each of the reviewer's points.
> * **Working with Trained Multimodal Classifier:**
> Our work focuses on robust multimodal learning under missing modality scenarios. We adapt pretrained unimodal encoders for feature extraction and train a classifier head to make predictions. The proposed CMPT method is not tied to any specific classifier. If a multimodal classifier is already trained, we can fine-tune it along with the encoders using CMPT to make it robust to missing modalities.
>
> * **Bigger Architecture and SigLIP model:**
> We thank the reviewer for raising this valuable observation. Below, we address each part of the comment:
>     * **Bigger Architecture:** As described in Section 3.1, the proposed method assumes the use of transformer-based pretrained unimodal encoders. We used ViT-B (from CLIP) due to its widespread use and to match the parameter scale of existing baselines and ensure fair comparisons. Our method can be applied to larger architectures, such as ViT-L or ViT-H, as long as they provide the embeddings equivalent to CLS tokens.
>     * **Compatibility with SigLIP Model:** To the best of our understanding, SigLIP [B] uses the same architecture as CLIP, but introduces a different loss function (pairwise Sigmoid loss) for pretraining. Since the model architecture remains the same, our method should work with SigLIP as well.
>     * **Models without CLS Tokens:** We use the CLS token as the embedding/feature representation of the input modality. If a model does not provide an explicit CLS token, we can (in principle) replace it with a global average-pooled token or some other representative feature. While building such models can be an interesting future research direction, our paper primarily focuses on contemporary transformer architectures that readily provide CLS tokens.
>
> * **Discussing MoRA in Related Work:**
> We thank the reviewer for bringing MoRA [A] to our attention. We have included a discussion of MoRA in the revised manuscript under the Related Work section.
>
> * **Reason Behind Choosing LoRA:**
> We chose LoRA for its simplicity, maturity, and widespread adoption. Our goal was to demonstrate that CMPT can achieve strong performance with minimal computational overhead. **As shown in Appendix A4, a LoRA with rank 1 was sufficient to deliver strong results in both complete and missing modality scenarios across multiple datasets. It also outperforms existing baseline models, as discussed in Section 4.4.** CMPT is not tied to any specific fine-tuning strategy and can be used with other low-rank adaptation methods such as DoRA or VeRA.
>
> * **Scalability to more modalities:**
> As discussed in **Section 5 (Limitations and Future Directions)**, this paper focuses on two-modality settings, primarily due to the availability of standardized benchmarks and established baselines that allow for fair and direct comparisons. We agree that evaluating CMPT in higher-order settings (with >2 modalities) is a valuable direction to assess its scalability and broader applicability. We are currently developing a general framework for this purpose, but consider such an extension beyond the scope of the current paper. We plan to present these results in future work.
>
> [A] Shi, Zhiyi, et al. "Mora: Lora guided multi-modal disease diagnosis with missing modality." International Conference on Medical Image Computing and Computer-Assisted Intervention. Cham: Springer Nature Switzerland, 2024.
>
> [B] Zhai, Xiaohua, et al. "Sigmoid loss for language image pre-training." Proceedings of the IEEE/CVF international conference on computer vision. 2023.

---

> > ### Author Response · Authors · 2025-08-06
> > **Response to the Comments from Reviewer ZCWM (2/2)**
> >
> > * **More Experiments:**
> > We thank the reviewer for acknowledging the simplicity of our model and believe that our current experimental evaluation is extensive and sufficient to support our claims. The key contribution of our work is achieving significant improvement and state-of-the-art performance with a simple and efficient method. Our empirical validation includes:
> >     * Experiments on five diverse multimodal datasets spanning different domains (image-text and audio-video).
> >     * Performance comparisons against 12 recent baselines across four multimodal datasets **under various missing modality scenarios (Section 4.3)**.
> >     * **Evaluation on complete modality scenarios** demonstrating that our method outperforms or matches existing baselines across five datasets **(Section 4.4)**.
> >     * Study in **generalization to unseen missing modality rates during inference**, where our method shows superior robustness **(Section 4.5)**.
> >     * Ablation studies analyze **CMPT effectiveness under different missing scenarios (Section 4.6.1 and Appendix A6)** and **per-class performance impact (Section 4.6.2 and Appendix A7)**.
> >     * Additional ablations for **optimal alignment loss weight (Section 4.6.3)** and **LoRA rank (Appendix A4)**.
> >     * **Qualitative analyses including t-SNE visualizations and attention maps** to provide deeper insight into the model’s behavior and failure cases **(Section 4.6.4 and Appendix A8)**.
> >
> >
> >     While additional experiments are always possible, we believe this comprehensive evaluation provides strong evidence of the effectiveness and adaptability of our framework. If the reviewer has some specific experiments in mind, we would be happy to consider them.
> >
> > We have revised the manuscript to incorporate the reviewer’s valuable feedback. **An annotated version highlighting all changes has been included in the supplementary materials.** We believe these revisions have significantly improved the clarity and overall quality of the paper, and we hope they adequately address all concerns. We sincerely thank the reviewer once again for their constructive comments.

---

> ### Comment · Action_Editor_rzFx · 2025-09-04
>
> Dear reviewer
>
> Please submit your final recommendation in light of the rebuttal and other reviews

---

> ### Comment · Reviewer_ZCWM · 2025-09-06
> **Response to the authors**
>
> Thank you for the reply and for the explanations.
> However, I still do have some concerns:
>
> - I still do believe that *performing experiments with larger backbone is valuable*, given the new trend of scaling the models even further.
> -  SigLIP models use GAP (Global Average Pooling) to obtain the embeddings, therefore, this method cannot be applied to this specific model.
> - Experiments with different low-rank adaptation methods could also be valuable for the community to show the behavior of the method.
>
> In the end, given the limited novelty (well-established methods applied), I do not believe that the current experiments are extensive enough, since one relatively small model is used (without knowing the method behavior on bigger models) and one low-rank method.

---

> ### Author Response · Authors · 2025-09-15
> **Experiments with larger backbone, SigLIP, and different low-rank adaptation methods (1/3)**
>
> We thank the reviewer for their detailed feedback and suggestions. In response to the concerns about model scaling, applicability to SigLIP, and other parameter-efficient adaptation methods, we have conducted additional experiments and added the results into the revised manuscript. Below, we address each of the reviewer's points.
>
> **1. Experiments with larger backbones:** To assess the scalability of our framework with larger encoders, we replaced the ViT-Base and BERT-Base encoders used in our primary experiments with ViT-Large and BERT-Large. We train the models on modality-complete data and evaluate them on both modality-complete and modality-incomplete scenarios on the UPMC Food-101 and MM-IMDb datasets.
>
> **R1T1: Experiments with larger backbones.**
>
> | Datasets | Available Modality | Missing Modality | ViT-Base and BERT-Base Params (M) | ViT-Base and BERT-Base Performance | ViT-Large and BERT-Large Params (M) | ViT-Large and BERT-Large Performance |
> |:---:|:---:|:---:|:---:|:---:|:---:|:---:|
> | MM-IMDB (F1-Macro) | Image | Text | 0.211 | 46.97 | 0.540 | **53.42** |
> | | Text | Image | | 56.32 | | **59.92** |
> | | Image - Text | - | | 63.58 | | **64.00** |
> | UPMC Food-101 (Accuracy) | Image | Text | 0.271 | 75.66 | 0.501 | **77.17** |
> | | Text | Image | | **85.31** | | 83.91 |
> | | Image - Text | - | | **94.47** | | 93.74 |
>
> The results, summarized in Table R1T1, show that our framework can leverage the increased capacity of larger backbones. On MM-IMDb, scaling yields consistent improvements in F1-Macro across all scenarios. The gains are particularly notable in missing-modality cases, with a +6.45% increase when text modality is missing and +3.60% when image modality is missing. These results suggest that stronger backbone representations improve CMPT’s ability to reconstruct missing features and maintain robust predictions. On UPMC Food-101, the larger backbone improves image-only accuracy by +1.51% but shows a decline of -1.41% for text-only performance. With both modalities available, performance remains comparable (94.47 vs. 93.74). This indicates a potential trade-off, where the added capacity may overfit to the text modality, reducing generalization when the image is absent.
>
> **In summary, CMPT scales effectively with larger backbones, particularly on MM-IMDb dataset. At the same time, the results highlight that the benefits of scaling depend on dataset and modality characteristics, underscoring the need for careful backbone selection to balance capacity and generalization. We have added Section A12 in the updated manuscript where we discuss these results.**

---

> > ### Author Response · Authors · 2025-09-15
> > **Experiments with larger backbone, SigLIP, and different low-rank adaptation methods (2/3)**
> >
> > **2. Experiments with SigLIP models:** To assess the architectural flexibility of our method, we replaced the ViT encoder with a pretrained SigLIP model [3], which employs Global Attention Pooling (GAP) instead of a CLS token. We align the CMPTs to the pooled token in the same way as we did with the CLS token. For audio–video datasets, we retained the AST model as the audio encoder, and for image–text datasets, we used BERT as the text encoder. In this setup, CMPT tokens must perform cross-architectural reconstruction: inferring missing audio or text features from SigLIP’s vision embeddings, and vice versa.
> >
> > **R1T2: Experiments with SigLIP models.**
> >
> > | Datasets | Available Modality | Missing Modality | ViT Accuracy (%) | SigLIP Accuracy (%) |
> > |:---:|:---:|:---:|:---:|:---:|
> > | UPMC Food-101 | Image | Text | 75.66 | **77.50** |
> > | | Text | Image | 85.31 | **85.58** |
> > | | Image - Text | - | 94.47 | **94.58** |
> > | AVE | Audio | Video | 85.21 | **86.32** |
> > | | Video | Audio | **84.58** | 30.60 |
> > | | Audio - Video | - | **96.77** | 94.78 |
> > | CREMA-D | Audio | Video | **67.20** | 65.73 |
> > | | Video | Audio | **76.21** | 35.35 |
> > | | Audio - Video | - | **88.84** | 84.68 |
> > | KS | Audio | Video | 68.27 | **69.66** |
> > | | Video | Audio | **85.77** | 30.76 |
> > | | Audio - Video | - | **91.21** | 89.44 |
> >
> > We train the models on modality-complete data and evaluate them on both modality-complete and modality-incomplete scenarios. The results in Table R1T2 show that our method is not restricted to a single encoder type. On the UPMC Food-101 dataset, the SigLIP-based model consistently outperforms the ViT-based model, demonstrating that CMPTs can be effectively applied to GAP-based vision backbones for vision–language tasks. In contrast, for audio–video datasets, SigLIP exhibits sharp performance degradation when audio is missing. Here, CMPTs struggle to recover audio features from global video embeddings, likely because GAP discards the fine-grained temporal information that is critical for inferring a corresponding audio representation. The reverse process, however, is more successful. When the video modality is missing, AST’s audio features provide a strong basis for reconstructing SigLIP’s global video representation. This suggests that AST’s feature space is more compatible with SigLIP than vice versa.
> >
> > **Overall, these findings indicate that CMPTs remain applicable to GAP-based models like SigLIP, but their effectiveness depends on the modality.** While GAP facilitates efficient vision embeddings, it limits the representational richness required for audio reconstruction, constraining CMPTs in certain cases. Nonetheless, **the results confirm that CMPTs are not inherently incompatible with SigLIP, though their utility is shaped by architectural and modality-specific factors. We have added Section A11 in the updated manuscript where we discuss these findings.**

---

> > > ### Author Response · Authors · 2025-09-15
> > > **Experiments with larger backbone, SigLIP, and different low-rank adaptation methods (3/3)**
> > >
> > > **3. Generalizability to other parameter-efficient adaptation methods:** To demonstrate that CMPTs are not restricted to a particular parameter-efficient adaptation method, we extended our analysis beyond LoRA, which was used in our primary experiments. **Specifically, we evaluated CMPTs with two recent alternatives: DoRA [1] and VeRA [2].** For a fair comparison, we fixed the rank to 1 across all methods. Models were trained on modality-complete data and evaluated on both modality-complete and modality-incomplete scenarios. The results are summarized in Table R1T3.
> > >
> > > **R1T3: Experiments with different parameter-efficient adaptation methods.**
> > >
> > > | Datasets | Available Modality | Missing Modality | LoRA Params (M) | LoRA Accuracy (%) | DoRA Params (M) | DoRA Accuracy (%) | VeRA Params (M) | VeRA Accuracy (%) |
> > > |:---:|:---:|:---:|:---:|:---:|:---:|:---:|:---:|:---:|
> > > | UPMC Food-101 | Image | Text | 0.271 | **75.66** | 0.302 | 69.03 | 0.155 | 70.82 |
> > > | | Text | Image | | **85.31** | | 83.52 | | 82.85 |
> > > | | Image - Text | - | | **94.47** | | 92.59 | | 92.34 |
> > > | AVE | Audio | Video | 0.173 | 85.21 | 0.246 | 86.57 | 0.098 | **89.05** |
> > > | | Video | Audio | | 84.58 | | **91.79** | | 90.05 |
> > > | | Audio - Video | - | | **96.77** | | 96.27 | | 96.27 |
> > > | CREMA-D | Audio | Video | 0.157 | **67.20** | 0.229 | 67.07 | 0.082 | 66.13 |
> > > | | Video | Audio | | **76.21** | | 73.12 | | 71.64 |
> > > | | Audio - Video | - | | **88.84** | | 88.04 | | 86.02 |
> > > | KS | Audio | Video | 0.176 | 68.27 | 0.248 | 68.77 | 0.101 | **69.93** |
> > > | | Video | Audio | | 85.77 | | **85.97** | | 83.69 |
> > > | | Audio - Video | - | | 91.21 | | **91.32** | | 90.71 |
> > >
> > > We find that the number of learnable parameters differs across methods, with VeRA being the most parameter-efficient and DoRA the least. In terms of performance, no single method consistently outperforms the others across all datasets. LoRA achieves the best overall results on the UPMC Food-101 and CREMA-D datasets, while DoRA shows superior performance on the Kinetics-Sound dataset and remains highly competitive on AVE. VeRA, while using substantially fewer parameters, delivers strong performance but is generally slightly below LoRA and DoRA, suggesting a trade-off between parameter efficiency and maximal performance.
> > >
> > > **Most importantly, our framework maintains robust performance in missing-modality scenarios regardless of the underlying adaptation method. The relative stability observed between complete and missing-modality settings is consistent across all three adaptation techniques.** These findings confirm that CMPTs function as a modular approach whose effectiveness is not tied to any specific adaptation strategy, underscoring their scalability and versatility across different parameter-efficient methods. We discuss these findings in **Section A10** of the updated manuscript.
> > >
> > > We appreciate the reviewer’s thoughtful feedback, which motivated us to broaden our study in several important directions. By incorporating experiments with larger backbones, GAP-based SigLIP models, and multiple parameter-efficient adaptation techniques, we demonstrate that CMPTs are scalable, versatile across architectures, and robust to the choice of adaptation method. We believe the revised manuscript now provides a comprehensive validation of our approach and directly addresses the reviewer’s concerns. **An annotated manuscript with all changes highlighted is included in the supplementary materials.** We sincerely thank the reviewer once again for their constructive comments.
> > >
> > >
> > > **References**
> > >
> > > [1] Liu, Shih-Yang, et al. "Dora: Weight-decomposed low-rank adaptation." Forty-first International Conference on Machine Learning. 2024.
> > >
> > > [2] Kopiczko, Dawid J., Tijmen Blankevoort, and Yuki M. Asano. "Vera: Vector-based random matrix adaptation." arXiv preprint arXiv:2310.11454 (2023).
> > >
> > > [3] Zhai, Xiaohua, et al. "Sigmoid loss for language image pre-training." Proceedings of the IEEE/CVF international conference on computer vision. 2023.

---

### Author Response · Authors · 2025-08-16
**follow up discussion**

Dear Reviewers and Editors,

We have submitted our response to all the reviewers comments and revised the papers accordingly.

We look forward to receiving any follow up questions and comments.

Thanks!

---

> ### Comment · Reviewer_SQkA · 2025-08-25
>
> I am not seeing the responses to my reviews and Reviewer Th7k's reviews. Could the authors kindly give me a pointer?

---

> ### Comment · Action_Editor_rzFx · 2025-08-25
>
> Dear reviewer
>
> You need to be logged into OpenReview to see the replies.

---

> ### Author Response · Authors · 2025-08-25
> **response visibility**
>
> Dear Reviewer SQkA,
>
> Thank you for bringing it up. We just noticed that our responses were only visible to editors and authors, which was not our intention. We are not sure why that happened and apologize for the inconvenience. The responses should be visible to everyone now.
>
> We look forward to hear your feedback and any other question you may have.
>
> Thanks!

---

### Decision · Action_Editor_rzFx · 2025-10-16

**Recommendation:** Accept with minor revision

**Additional Comments:**

Thank you for updating the paper during the rebuttal period with the reviewers' comments. Please add Github links as promised in the final version.

**Audience:**

Yes

**Audience Explanation:**

The authors propose to enhance the robustness of multimodal models in the scenario where a modality is missing. The authors do so by introducing Cross-Model Proxy Tokens which are supervised with an alignment loss.

The reviewers all agreed that the method is simple yet effective method that shows good results on a number of benchmarks. Therefore, it will be of interest to members of TMLR's audience.

That said, reviewers felt that the novelty is limited and the scope too narrow -- the work is limited to two modalities and a single adaptation method. Addressing these limitations would make the paper of interest to a wider range of the TMLR audience.

**Claims And Evidence:**

Yes

**Claims Explanation:**

The contributions of the paper are experimental, and supported sufficiently with experiments in the main paper and also in the rebuttal. Reviewers were unanimous that the claims are supported adequately.